# Beyond Problem Solving: UOJ-Bench for Evaluating Code Generation, Hacking, and Repair in Competitive Programming

**Tingqiang Xu** [* 1 2]   **Hangrui Zhou** [* 1 2]   **Tianle Cai** [3]   **Alex Gu** [4]   **Kaifeng Lyu** [1 2]

## Abstract

Despite strong performance in competitive programming, the role of Large Language Models (LLMs) in supporting human learning in the same setting remains largely unexplored. In this work, we introduce **UOJ-Bench**, a benchmark designed to evaluate not only the problem-solving ability of LLMs, but also their ability to identify errors in human-written code—a crucial educational activity traditionally supported by running test cases over online judge systems. UOJ-Bench consists of three distinct tasks: code generation, code hacking, and code repair, all constructed from real-world code submissions on the Universal Online Judge (UOJ)[1] and evaluated through UOJ's native judging infrastructure. Our results show that under one-shot evaluation, even the strongest models fail to identify errors in more than 50% of a set of submissions that have been found to be incorrect by UOJ users. While test-time scaling improves success rates to above 90%, the substantial computational costs incurred from model inference limit its practicality for large-scale deployment. Despite these limitations, we find that the best-performing models under test-time scaling can uncover errors in over 5% of full-score submissions across around 30 problems, suggesting that frontier LLMs can already provide complementary signals beyond standard judging systems. UOJ-Bench is publicly available at https://github.com/hehezhou/UOJ-Bench.

## 1. Introduction

Large Language Models (LLMs) have demonstrated remarkable proficiency in problem solving, especially in the domain of competitive programming (CP). OpenAI o3 (OpenAI, 2025) and Gemini 2.5 Deep Think (Gemini Team, 2025; Lin & Cheng, 2025) were reported to achieve gold-medal level performance at the 2024 International Olympiad in Informatics (IOI) and the 2025 International Collegiate Programming Contest (ICPC) World Finals, respectively. Consequently, a growing number of benchmarks (Jain et al., 2025; Zheng et al., 2025) have emerged, largely centered on evaluating LLMs' autonomous ability to generate correct solutions from problem descriptions.

However, CP problems are not designed to benchmark machines, but to cultivate human capabilities such as problem solving, resilience, teamwork, as well as a distinctive mode of thinking, known as computational thinking (Wing, 2006; Yuen et al., 2023). In this spirit, we take a complementary perspective and focus on evaluating LLMs' capabilities in supporting *human learning* in competitive programming. In particular, we aim to understand: *to what extent can current LLMs provide additional value beyond the existing software infrastructure in competitive programming?*

**Beyond Problem Solving.**   A key capability of LLMs that may help improve students' learning in competitive programming is the ability to identify errors in human-written code. Traditionally, checking the correctness of a code solution is supported by online judge (OJ) systems that maintain a fixed set of test cases for each problem. In typical practice, students first write a solution and submit it to an OJ, which then evaluates the code against these test cases and returns a score. If the score is not full, students may iteratively refine and resubmit their code until it passes all test cases.

While OJs are effective in many cases, they suffer from the following two limitations. First, the standard test cases are static and are insufficient to detect all incorrect solutions. Although these test cases are carefully crafted by problem setters to expose common errors, students often approach the same problem in highly creative ways, which can lead to subtle and unexpected mistakes. As a result, it is not uncommon for a solution to pass all standard test cases

---

[*]Equal contribution   [1]Institute for Interdisciplinary Information Sciences, Tsinghua University   [2]Universal Online Judge   [3]ByteDance Seed   [4]Massachusetts Institute of Technology. Correspondence to: Kaifeng Lyu <klyu@mail.tsinghua.edu.cn>.

*Proceedings of the 43rd International Conference on Machine Learning*, Seoul, South Korea. PMLR 306, 2026. Copyright 2026 by the author(s).

[1]Universal Online Judge: https://uoj.ac

*Table 1.* Comparison of UOJ-Bench with existing code reasoning benchmarks. ✄ in the Repair column indicates that the benchmark is restricted to repairing code generated by LLMs (self-repair). ✄ in Time Limit Checking denotes that the benchmark uses a custom or surrogate judging environment rather than the original judging infrastructure. Detailed statistics on the number of problems and task instances are provided in Table 5.

| Competitive Programming Benchmarks | Problems | | Tasks | | | Judging Support | |
|---|---|---|---|---|---|---|---|
| | IOI-level | UOJ Originals | Generation | Hacking | Repair | Non-Trivial Grading | Time Limit Checking |
| LiveCodeBench (Jain et al., 2025) | × | × | ✓ | × | ✄ | × | × |
| LiveCodeBench Pro (Zheng et al., 2025) | ✓ | × | ✓ | × | × | × | ✄ |
| CodeELO (Quan et al., 2025) | ✓ | × | ✓ | × | × | ✓ | ✓ |
| REFUTE (Sinha et al., 2025) | ✓ | × | × | ✓ | × | × | × |
| ELABORATION (Yang et al., 2025a) | × | × | ✓ | × | ✄ | × | × |
| UOJ Bench (**Ours**) | ✓ | ✓ | ✓ | ✓ | ✓ | ✓ | ✓ |

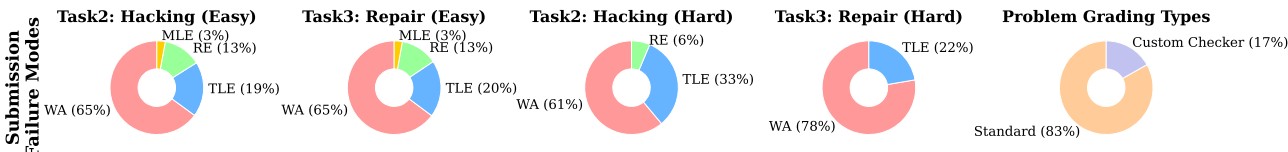

*Figure 1.* Detailed statistics of the UOJ Bench dataset. WA, TLE, RE, and MLE denote Wrong Answer, Time Limit Exceeded, Runtime Error, and Memory Limit Exceeded verdicts, respectively.

while still being incorrect upon closer examination. In this paper, we refer to errors that can be exposed by standard test cases as *overt errors*, and those that cannot be exposed by standard test cases as *covert errors*.

Second, current OJs provide only numerical scores or coarse verdicts, such as "Wrong Answer" (WA) or "Time Limit Exceeded" (TLE). Such feedback offers little guidance when a solution is incorrect, often leaving students with a lengthy debugging process to identify and fix the underlying issues.

These two limitations motivate us to evaluate LLMs through two code reasoning tasks:

- **Code Hacking:** Given a CP problem and a buggy solution, generate a test case to break the solution;

- **Code Repair:** Given a CP problem and a buggy solution, generate a minimal patch that fixes the code so that it passes all test cases.

LLMs that perform well on code hacking could help online judges automatically generate test cases to uncover covert errors, while strong performance on code repair could enable more detailed, step-by-step feedback to guide students toward correct solutions. Together, these capabilities could democratize programming education by making high-quality feedback accessible to more learners.

**UOJ-Bench.** In collaboration with the Universal Online Judge (UOJ), a popular community-driven online judge system in China, we introduce **UOJ-Bench**, a challenging benchmark that evaluates the capabilities of LLMs ranging from **code generation** to **code hacking** and **code repair** in competitive programming with real-world data. UOJ-

Bench places particular emphasis on the latter two tasks, **code hacking** and **code repair**, for their educational value, while retaining **code generation** to assess models' ability to solve the underlying CP problems.

To measure the marginal value LLMs provide beyond standard test cases, we design two difficulty levels:

- **Easy:** For overt errors that can be exposed by human-crafted standard test cases, are LLMs also able to identify and fix these errors?

- **Hard:** Are LLMs able to identify and fix covert errors that are beyond what standard test cases can reveal?

To construct the Easy level of the hacking and repair tasks, we collect 500 pairs of code submissions that initially receive partial scores and later obtain full scores after being revised by the original authors. While this process could, in principle, be carried out on any online judge system that publicizes user submissions, UOJ provides a uniquely suitable testing ground for the Hard level tasks due to its distinctive **Hack mechanism**. This mechanism consolidates community efforts to identify covert errors in code submissions by allowing users to submit test cases designed to break other users' solutions that have already passed all existing test cases in the system. Although detecting covert errors typically requires substantial human effort, UOJ has accumulated over 2,000 successful hacks over the years, providing a rich data source for constructing the Hard level tasks. See Section 3.1 for more details.

**Zero-Day Hacking.** Beyond the Easy and Hard levels, we further introduce an extended evaluation set for code hack-

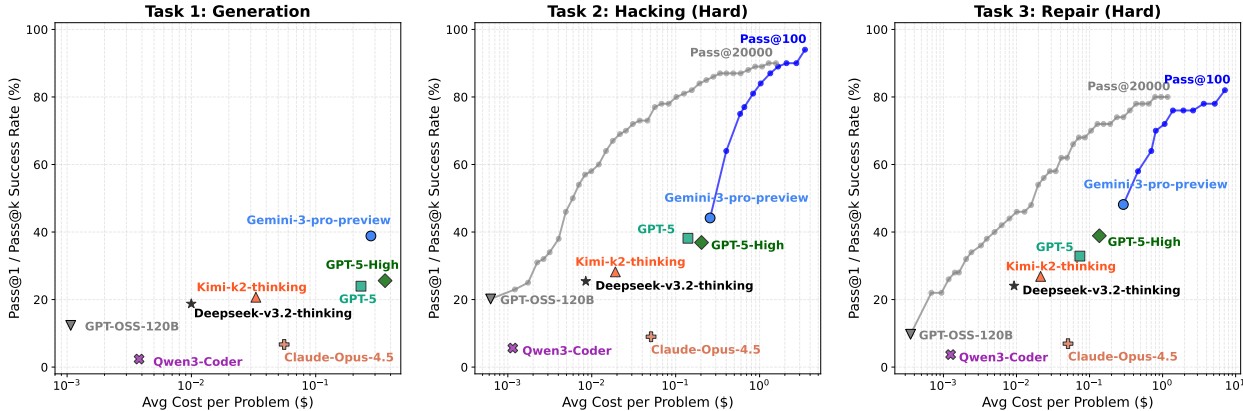

*Figure 2.* Cost-performance analysis across three tasks: (a) Generation, (b) Hacking (Hard), and (c) Repair (Hard). The plots compare success rates (Pass@1) against the average inference cost per problem. For GPT-OSS-120B and Gemini-3-pro-preview, we additionally illustrate test-time scaling capability by plotting **Pass@**$k$ curves (up to $k = 20,000$ and $k = 100$, respectively) on randomly selected subsets (100 samples for Task 2, 50 for Task 3). Gemini-3-pro-preview demonstrates the strongest overall performance, while GPT-OSS-120B offers the best cost-effectiveness.

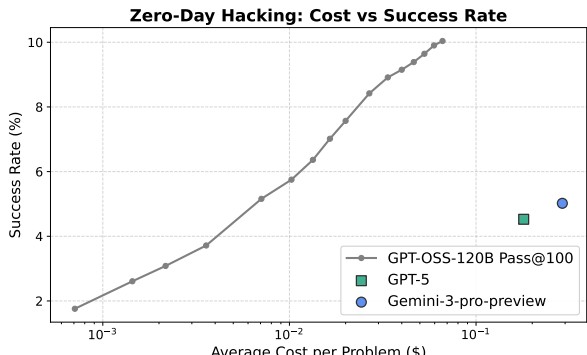

*Figure 3.* Cost-performance analysis for Zero-Day Hacking. GPT-OSS-120B under test-time scaling is the most cost-effective.

ing, **Zero-Day-Hacking-5K**, which contains 5,060 sampled full-score submissions. Unlike the Hard set, where covert errors have already been exposed by human-generated hacks, this set contains submissions with no known failing test cases at the time of collection. This setting therefore evaluates whether LLMs can discover new covert errors that have not yet been discovered, analogous to identifying "zero-day" vulnerabilities in software security. Because such errors can be rare under uniform sampling, we construct this set to be more likely to contain yet-undiscovered vulnerabilities: submissions are sampled from different problems in proportion to the historical number of successful hacks per problem.

**Key Features.** Based on the Hack mechanism and other unique features of UOJ, **UOJ-Bench** offers the following key characteristics:

1. UOJ-Bench provides a comprehensive evaluation of LLMs' capabilities in code generation, hacking, and repair in competitive programming, with an explicit split between overt and covert errors.

2. The use of open-access user code submissions ensures that the code hacking and repair tasks are directly connected to the performance of LLMs in the real world.

3. The collaboration with UOJ enables us to evaluate LLM-generated code, hacks, and patches directly through UOJ's internal API. Compared to existing benchmarks that often build their own grading system that gives up problems with non-trivial grading and strict time limit checking, we report more thorough results with no gap to the real-world testing environment.

**Key Findings.** By evaluating both proprietary models and open-source models on UOJ-Bench, we find that:

- All models perform with success rate below 60% on the Easy level of the hacking and repair tasks and below 50% on the Hard level. While test-time scaling improves success rates to above 90%, the substantial computational costs incurred from model inference its practicality for large-scale deployment. This indicates that current LLMs have ample room for improvement in identifying and fixing errors for educational purposes.

- While proprietary models generally perform better than open-source models, under test-time scaling, the most cost-effective model is GPT-OSS-120B, which beats Gemini 3 Pro Preview by a significant margin under the same cost.

- Surprisingly, GPT-OSS-120B under test-time scaling is able to uncover errors in over 10% of all submissions in Zero-Day-Hacking-5K and over 5% of full-score submissions across around 30 problems, which suggests that frontier LLMs can already provide complementary signals beyond standard judging systems.

**Outline.** The rest of the paper is organized as follows. We provide a detailed description of UOJ-Bench in Section 3. We then present the experimental setup and main results in Section 4. The setup and results of the zero-day hacking evaluation are presented in Section 5. Finally, we conclude in Section 6.

## 2. Related Works

**Competitive Programming Benchmarks.** Competitive programming (CP) is a challenging test-bed for assessing code reasoning. Since the release of CodeContests (Li et al., 2022), the field has evolved rapidly to challenge advanced models. Efforts to refine data quality and scale include CodeContests$^+$ (Wang et al., 2025b), TACO (Li et al., 2023), OJBench (Wang et al., 2025c), and AetherCode (Wang et al., 2025a). To target specific high-difficulty domains, USACO Bench (Shi et al.) and ICPC-Eval (Xu et al., 2026) introduce Olympiad-level constraints. Furthermore, to mitigate data contamination, dynamic and holistic evaluation frameworks such as LiveCodeBench (Jain et al., 2025) and CodeElo (Quan et al., 2025) emulate real-world contest environments. However, these benchmarks predominantly focus on *code generation*, while UOJ-Bench extends the code reasoning tests to *code hacking* and *repair*. Moreover, most of these benchmarks rely on custom or surrogate judging pipelines, which may introduce discrepancies from the original online judge infrastructure. In contrast, UOJ-Bench uses UOJ's native judging infrastructure. Among existing benchmarks, only CodeElo (Quan et al., 2025) similarly evaluates submissions through Codeforces' native judging infrastructure.

**Test Case Generation and Code Hacking.** Beyond code synthesis, evaluating LLMs as "testers" is gaining traction. Research spans from generating standard tests for coverage (Zhou et al., 2025; Fu et al., 2025; He et al., 2025) to more rigorous assessments of fault exposure (Yang et al., 2025b) and reliable generator creation (Cao et al., 2026). This adversarial capability is also explored in domain-specific fuzzing for security (Zhang et al., 2023), deep learning libraries (Deng et al., 2023), and compilers (Ye et al., 2021). In the CP domain, REFUTE (Sinha et al., 2025) evaluates LLMs' code hacking ability but focuses primarily on overt errors that fail standard system tests, and our analysis finds that their benchmark only contains 42 samples with covert errors. UOJ-Bench addresses this limitation by providing a substantially larger set of submissions with covert errors. Beyond scale, UOJ-Bench covers additional failure modes, including efficiency issues such as TLE, and introduces a zero-day hacking setting for discovering previously unknown bugs, while also incorporating a Code Repair task to assess the full debugging loop.

**Automated Program Repair Benchmarks.** Recent automated program repair benchmarks address challenges at varying scales, ranging from repository-level issue resolution in SWE-bench (Jimenez et al., 2024) to fine-grained algorithmic repair. However, existing algorithmic datasets exhibit two primary limitations concerning the source of errors and problem complexity. First, benchmarks like LiveCodeBench (Jain et al., 2025) and ELABORATION (Yang et al., 2025a) primarily evaluate "self-repair," where models must refine their own generated output. This setup obscures the distinct challenge of fixing human-written code, as model-generated errors may follow different distributions from human-written bugs. Second, benchmarks that do target human-written code, such as RunBugRun (Prenner & Robbes, 2023) and RealHumanEval (Mozannar et al., 2025), generally focus on elementary-level problems. Run-BugRun is limited to small-scale solutions with standard logical faults (e.g., sorting or string manipulation), while RealHumanEval targets interview-style queries. In contrast, UOJ-Bench bridges this gap by targeting high-complexity, IOI-level algorithms where human solutions pass initial tests but fail under adversarial hacking, requiring deep algorithmic reasoning rather than fixes to relatively simple faults.

## 3. UOJ-Bench

Building upon real-world data from UOJ, we construct a code reasoning benchmark that comprehensively evaluates the capabilities of LLMs in three distinct tasks in the context of competitive programming: **code generation** (Task 1), **code hacking** (Task 2), and **code repair** (Task 3).

### 3.1. What is UOJ?

The Universal Online Judge is a community-driven online-judge platform widely used by competitive programmers in China. It distinguishes itself through several core features that make it an ideal source for rigorous benchmarking.

**Comprehensive Problem Repository.** UOJ hosts a vast archive of advanced algorithmic problems. In addition to community-authored content, the repository aggregates problems from prestigious national and international competitions. These include tasks from the IOI, JOISC, and the full hierarchy of Chinese contests (NOI, NOIP, China Team Selection, and National Training Camps). These diverse sources inherently ensure a realistic distribution of difficulty and algorithmic variety.

**Expert Community and Original Contests.** The platform is maintained by a rotating committee of top-tier competitive programmers, typically selected from the Chinese National Training Team (the top 50 programmers nationwide each year). This team organizes high-quality original contests (e.g., UOJ Rounds) featuring problems known for

their extreme difficulty and depth, often matching or exceeding the standards of NOI and IOI. This ensures that the "UOJ Originals" portion of our benchmark represents the cutting edge of algorithmic reasoning.

**The Hack Mechanism.** A distinguishing feature of UOJ is its evolutionary test data system. For each problem, the test data consist of standard `Tests` and dynamic `Extra Tests`. For problems where hacking is enabled, typically excluding those with purely randomized inputs or prohibitive input validation costs, users can challenge accepted solutions by submitting a *hack*: an input file that contains a targeted counter-example designed to break the code. Each hack is **fully verifiable** because every hackable problem is equipped with a correct reference solution and an input validator. Concretely, a submitted hack input is first checked by the input validator, then executed on the reference solution to obtain the expected output, and finally run the targeted submission on the hack input and compare the output with the expected output. If the hack succeeds (i.e., the targeted submission fails to produce the expected output), the test case is automatically added to the `Extra Tests`, which then triggers a re-evaluation of all full-score submissions. This mechanism effectively crowdsources the discovery of edge cases and provides our benchmark with a rich repository of subtle bugs that standard tests often miss.

**Open-Access Submissions and Data.** Unlike platforms where users can only access their own submissions, UOJ adopts an open-access policy under which all user submissions and hack submissions, including incorrect ones, are accessible to the community. This transparency is crucial for our benchmark construction because it enables us to harvest pairs of (buggy code, hack input) and trace subsequent submissions that repair the buggy code.

**Support for Custom Checkers.** UOJ natively supports custom checkers for problems with multiple valid outputs, allowing the judge to validate semantic correctness rather than a strict text matching against a single fixed reference output. We refer to this capability as *non-trivial grading*. For example, if a problem asks the program to construct a graph with specific connectivity properties, the checker may parse the submitted graph and verify the required topology, instead of comparing the output to a fixed graph. This allows UOJ-Bench to include problems that benchmarks based on custom or surrogate judging pipelines may exclude due to verification complexity (e.g., REFUTE (Sinha et al., 2025); see Table 1). Detailed statistics are provided in Figure 1.

### 3.2. Data Curation

We prioritize problems with extensive community scrutiny to better evaluate code robustness and algorithmic reasoning. The data collection cutoff is **September 15, 2025**.

**Code Hacking Data (Hard)** The "Hard" subset of the hacking data is derived from successful community hack attempts recorded on UOJ. These instances represent counter-examples where a submission passed the standard test cases but failed against a user-submitted test case. To ensure reproducibility and standardization, we apply a rigorous filtering pipeline (detailed in Appendix B). Briefly, this process excludes non-standard problem formats, randomization-dependent hacks, "self-hacks", and outdated submissions that no longer fail under current UOJ hardware conditions.

To maintain dataset balance, we select a maximum of ten distinct hack examples per problem, covering diverse failure modes including Wrong Answer (WA), Runtime Error (RE), and Time Limit Exceeded (TLE).

**Code Repair Data (Hard)** For each valid hacked submission identified above, we search for subsequent attempts by the same user to find a fix. We construct (buggy code, fixed code) pairs where the subsequent submission passes all test cases. To ensure the benchmark focuses on precise debugging rather than extensive rewriting, we enforce a similarity constraint: the similarity score $1 - d_{\text{edit}}/L$ must be at least **0.95**, where $d_{\text{edit}}$ is the Levenshtein edit distance and $L$ is the length of the longer code in the pair.

These candidate pairs undergo a joint classification and filtering process (assisted by an LLM) to assign each pair to one of the following error types: **Corner Case Error** (fails only on boundary conditions, such as overflow, precision, special cases); **Careless Error** (implementation oversights such as incorrect types, swapped variables, or initialization failures); and **Wrong-but-Works Algorithm** (the logic is fundamentally flawed but coincidentally passes most tests).

Simultaneously, we filter out pairs falling into the following exclusion categories: **Both-Wrong/Irrelevant** (the "fixed" code is incorrect or unrelated to the buggy version); **Hard-coded Fixes** (the fix simply hardcodes the output for the hack case without addressing the root cause); **Randomness Exploits** (the error or fix relates to exposing or patching random seeds rather than algorithmic logic). To validate the pipeline, we manually audited all submission pairs classified as "Both-Wrong" and "Wrong-but-Works" in the code repair (hard) dataset, as these represent the most challenging categories. Our review confirmed that the LLM achieved a 100% accuracy rate on these instances.

Finally, we apply the same re-submission check as in the Hacking task. We retain a maximum of five repair pairs per problem.

**Code Repair Data (Easy)** To complement the "Hard" data with *covert* errors, we construct an "Easy" subset with *overt* errors. We identify user submissions that achieved a partial score ($\geq 60$) but failed to pass all test cases. For these submissions, we locate a corresponding fixed version

by the same procedure described above. From a filtered pool of approximately 10,000 candidates, we randomly sample 500 pairs that pass the re-submission verification to form the "Easy" level of the Repair dataset.

**Code Hacking Data (Easy)** The "Easy" hacking dataset is derived from the same pool of 500 buggy submissions used for the Easy Repair task. However, since hacking is not enabled for all problems (e.g., due to missing validators or specific configuration constraints), we exclude 21 instances. Consequently, the final "Easy" hacking dataset consists of 479 valid buggy submissions.

**Code Generation Data** The generation dataset is composed of the union of problems selected for the Easy and Hard versions of the Hacking and Repair tasks, supplemented by all problems from UOJ original contests. This aggregation ensures a broad distribution across difficulty levels and algorithmic domains.

In all three tasks, all problem statements are translated from Chinese to English using an LLM to facilitate standardized benchmarking. In total, UOJ bench consists of 672 CP problems for Task 1, 479 and 1,046 code submissions for Task 2 (Easy and Hard), and 500 and 216 code submissions for Task 3 (Easy and Hard).

### 3.3. Composition

Unlike existing benchmarks such as REFUTE (Sinha et al., 2025), which primarily focus on logic errors leading to Wrong Answers (WA), UOJ-Bench covers a broader range of failure modes. Our dataset explicitly includes submissions that trigger **Runtime Errors (RE)**, **Time Limit Exceeded (TLE)**, and **Memory Limit Exceeded (MLE)** verdicts, providing a more complete view of real-world debugging scenarios.

Furthermore, we incorporate a significant proportion of problems requiring **custom checkers**. This allows us to include constructive algorithms and problems with multiple valid solutions, which are often excluded from other benchmarks due to the complexity of validation.

Our analysis in Appendix C demonstrates that these factors are critical bottlenecks: addressing efficiency constraints (TLE) and handling constructive ambiguity (Custom Checkers) prove more challenging for current models than fixing standard logical errors. Figure 1 details the distribution of these failure modes and problem types.

We also report the programming language distribution across our benchmark. In the field of competitive programming, C++ accounts for the vast majority of submissions due to strict execution speed constraints, and UOJ-Bench reflects this real-world prevalence. Within the Code Hacking dataset (Hard), 1,018 submissions are written in C++, 23 in

Pascal, and 4 in Python. The Code Repair dataset (Hard) follows a similar trend, containing 214 C++ submission pairs and 2 Pascal pairs. This language bias is consistent with existing literature; for example, REFUTE (Sinha et al., 2025) similarly comprises 317 C++ samples and 7 Python samples out of its 324 total instances.

### 3.4. Difficulty

We categorize problems into four difficulty levels: **Easy**, **Medium**, **Hard**, and **Ultrahard**. These difficulty levels are determined by synthesizing metrics from multiple sources. To contextualize the spectrum: the Easy level is analogous to the first problem in a Codeforces Div. 3 contest, while the Ultra Hard level is comparable to the last problem in the International Olympiad in Informatics (IOI). Further details are provided in Appendix B.

Figure 4 presents the distribution of problems across different sources and difficulty levels. The dataset covers a wide range of high-quality contests, categorized as follows: (1) **NOI Series:** major Chinese competitions such as NOI, NOIP, CSP, Winter Camp (WC), and provincial selections; (2) **National Team Training:** China Team Training (CTT), China Team Selection (CTS), Team Homework, and Mutual Tests; (3) **International Contests:** IOI, JOISC, and APIO; (4) **UOJ Originals:** original contests hosted on UOJ (e.g., UR, UER, ULR), along with additional original problems authored by UOJ administrators.

## 4. Evaluation

### 4.1. Experimental Setup

We conduct a comprehensive evaluation using a suite of state-of-the-art frontier models, including **Gemini-3-pro-preview**, **Deepseek-v3.2**, **Kimi-k2-thinking**, **GPT-5**, **GPT-OSS-120B**, **Qwen3-Coder**, and **Claude-Opus-4.5**. To handle complex reasoning chains and large code contexts, the maximum response length is set to the recommended upper bound (at least 64k tokens).

### 4.2. Results

In the Direct Evaluation setting, models perform tasks in a single-turn, zero-shot manner. The prompts are provided in Appendix A. For Task 1 (Generation) and Task 3 (Repair), the generated solutions are submitted directly to the UOJ evaluation pipeline using **UOJ's internal API**; success is defined as passing all primary `Tests` and `Extra Tests`. For Task 3 specifically, we enforce a pre-validation constraint: the generated patch must modify no more than 10% of the original code to be considered valid. For Task 2 (Hacking), the generated data synthesis script is submitted directly to UOJ to verify if it successfully triggers a failure

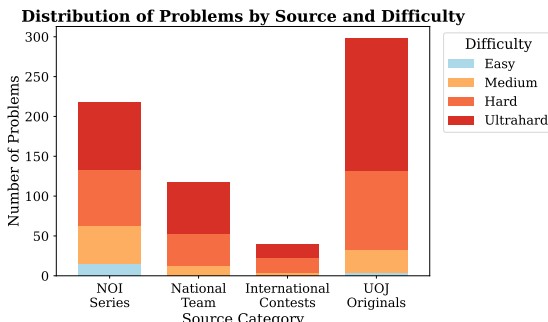

*Figure 4.* Distribution of Problems by Source and Difficulty.

*Table 2.* Task 1 (Generation) Performance by Difficulty (Pass@1)

| Model | Easy | Medium | Hard | Ultrahard | Overall |
|---|---|---|---|---|---|
| Gemini-3-pro-preview | 94.44 | 80.00 | 46.90 | 18.62 | 38.84 |
| GPT-5-high | 88.89 | 64.21 | 28.76 | 9.01 | 25.60 |
| GPT-5 | 77.78 | 63.16 | 25.66 | 8.71 | 23.96 |
| Kimi-k2-thinking | 88.89 | 55.79 | 23.89 | 4.80 | 20.68 |
| Deepseek-v3.2-thinking | 77.78 | 58.95 | 18.14 | 4.50 | 18.75 |
| GPT-OSS-120B | 77.78 | 40.00 | 11.50 | 1.50 | 12.35 |
| Claude-Opus-4-5 | 50.00 | 23.16 | 5.31 | 0.60 | 6.70 |
| Qwen3-Coder | 33.33 | 9.47 | 0.44 | 0.00 | 2.38 |

in the target solution (i.e., constitutes a valid hack). Across all tasks, performance is quantified using the **Pass@1** metric. Figure 2 shows the model performance and the costs.

**Task 1: Generation Performance** Table 2 summarizes code generation performance across difficulty levels. As task difficulty increases, model performance consistently degrades. Even the strongest model, Gemini-3-pro-preview, struggles with the most complex problems, achieving only a 38.84% overall score.

**Task 2 & 3: Hacking and Repair Performance** Table 3 reports results for the Hacking and Repair tasks under the direct prompting setting. Results show that models perform noticeably better at identifying and fixing overt errors than covert ones, and fail to detect errors in over 50% of submissions in the "Hard" partition of the Hacking and Repair dataset.

**Contamination Analysis.** To assess potential data contamination, we evaluate model performance relative to problem release time, focusing on *Hard* problems to control for difficulty shifts. We observe that *code generation* shows a clear performance decline on more recent problems, suggesting some reliance on memorized solutions. In contrast, the core tasks, *code hacking* and *code repair*, remain stable over time, indicating they are less affected by contamination and rely more on genuine reasoning and logic verification. Detailed methodology and temporal analyses are provided in Appendix C.

### 4.3. Test-Time Scaling

In practical deployment, large language models can be queried multiple times for the same task. This is especially natural for *code hacking* and *repair*, where a model can iteratively propose alternative attacks or fixes until one succeeds. Motivated by this usage pattern, we study the effect of test-time scaling by evaluating performance under multiple inference attempts.

We evaluate **Pass@**$k$ performance in Task 2 and Task 3. By generating $k$ independent candidates for each problem, we

*Table 3.* Direct Evaluation Results for Hack and Repair (Pass@1)

| Model | Hack (Easy) | Hack (Hard) | Repair (Easy) | Repair (Hard) |
|---|---|---|---|---|
| Gemini-3-pro-preview | 61.38 | 44.17 | 53.60 | 48.15 |
| GPT-5-high | 50.73 | 36.90 | 43.80 | 38.89 |
| GPT-5 | 51.77 | 38.15 | 41.40 | 32.87 |
| Kimi-k2-thinking | 45.93 | 28.20 | 32.20 | 26.85 |
| Deepseek-v3.2-thinking | 40.29 | 25.43 | 26.20 | 24.07 |
| GPT-OSS-120b | 31.11 | 20.17 | 20.20 | 9.72 |
| Claude-Opus-4-5 | 12.11 | 8.99 | 16.60 | 6.94 |
| Qwen3-Coder | 7.31 | 5.64 | 3.60 | 3.70 |

measure the probability that at least one solution is correct.

**Experimental Setup** To reduce API inference costs, we sampled a subset of 100 instances from the "hard" partition of the hacking dataset and 50 instances from the "hard" partition of the repair dataset to assess test-time scaling. We selected two models: GPT-OSS-120B, representing the most cost-efficient option, and Gemini-3-pro-preview, representing the strongest performance (see Figure 2). Due to these representative characteristics, we employ these same two models for all subsequent additional experiments.

The results are illustrated by the grey and blue dots in Figure 2. The corresponding plots with $k$ on the $x$-axis are provided in Appendix C. Under test-time scaling, Gemini-3-pro-preview demonstrates robust performance, successfully detecting 93 out of 100 hacking samples and repairing 41 out of 50 samples.

**Cost Analysis.** Inference costs are estimated using real-time pricing from **OpenRouter**, with results shown in Figure 2. In competitive programming, solution code and associated reasoning are often lengthy, leading to substantial token consumption per sample. As a result, brute-force test-time scaling encounters a severe economic bottleneck. As illustrated in Figure 2, the cost per successful solution increases with $k$; in practice, scaling beyond $k = 20,000$ costs approximately $1 per problem.

Given that UOJ processes roughly 100,000 submissions per year, evaluating submissions using LLM-based methods

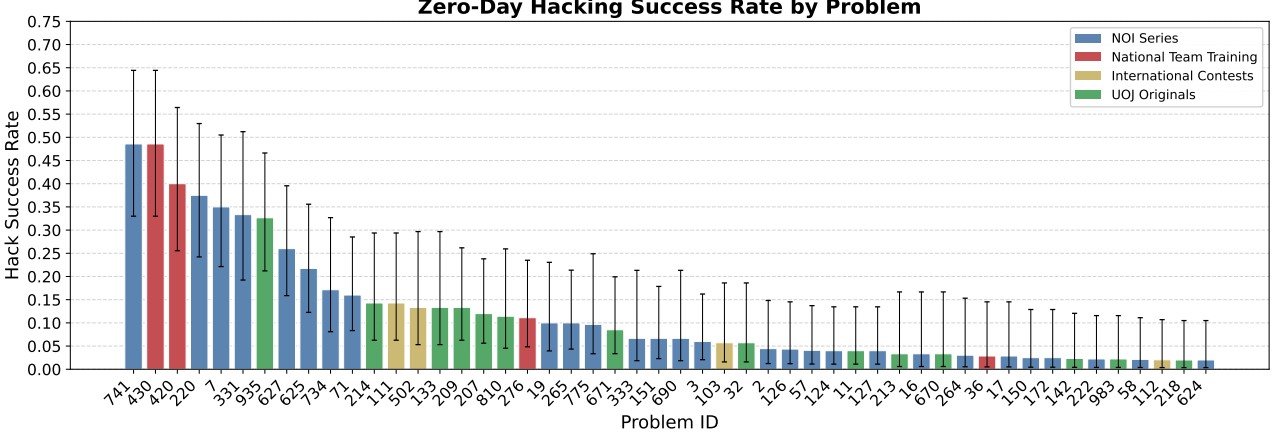

*Figure 5.* Zero-Day Hacking success rates across problems. The chart displays the percentage of target submissions (originally deemed Accepted) for which the models successfully generated a valid counter-example. The significant variance indicates that certain problems possess less robust historical test suites, resulting in a higher prevalence of latent errors. Error bars represent 90% confidence intervals.

would incur an annual cost on the order of $100,000—over two orders of magnitude higher than the cost of traditional CPU-based judging (under $500 per year). This cost disparity makes large-scale deployment of LLM-based judging economically unsustainable. Detailed token statistics and cost calculations are provided in Appendix B.

Finally, we manually inspected cases where test-time scaling failed to produce a hack. We found that these typically involve subtle logic errors requiring highly specific edge cases, or target solutions using randomized heuristics that make constructing counter-examples computationally difficult, even for expert programmers (see Section D for a concrete example).

**Agentic Evaluation for Hacking and Repair.** We compare direct sampling with an agentic evaluation framework for **Task 2 (Hacking)** and **Task 3 (Repair)**. Our analysis reveals a distinct divergence in optimization strategies: while Program Repair benefits substantially from iterative debugging, Hack Generation is more effectively optimized through diverse parallel sampling.

To implement this agentic approach, we employ a ReAct framework that engages in multiple interactive rounds. The feedback mechanisms for each task are structured as follows:

- **Task 2 (Hacking):** If the previous hack attempt fails, the agent receives specific feedback to guide its next attempt. If the failure is due to script compilation errors or invalid input formats, the error log is returned. If the test case is valid but the hack is unsuccessful (i.e., the target solution's output matches the reference solution), the agent is provided with the actual output produced by the target code. The agent is then prompted to generate a new, stronger counter-example.

- **Task 3 (Repair):** If the previous patch fails to apply, the specific error message is returned to the agent. If the patch is successfully applied but the modified code fails validation, the agent utilizes feedback from the online judge, such as compilation errors, runtime verdicts, or failed test case details, to iteratively diagnose the bug and refine the submission.

Comprehensive prompt templates are provided in Appendix A. Figure 6 illustrates the performance scaling of the Agentic setting versus the Direct baseline (Pass@k) over 10 turns. As shown in the left plot, the agentic hacking approach closely mirrors the baseline, suggesting that feedback offers little advantage over simple enumeration for this task. Conversely, the right plot demonstrates a steep upward trajectory for the repair task, confirming that iterative refinement significantly outperforms direct sampling.

**Additional Analysis** In Appendix C, we present additional experiments complementing the main results. We analyze model performance across difficulty levels, error types, grading mechanisms, and problem categories to reveal systematic debugging bottlenecks. We further study the relationships between generation, hacking, and repair, showing that debugging skills are partially decoupled from solution synthesis, and provide details on contamination analysis. Additional analyses examine the edit distance constraint, repair format (patch vs. full rewrite), prompt language (Chinese vs. English), test-time scaling, and educational validation.

## 5. Zero-Day Hacking

While the **Code Hacking** task (Task 2) evaluates models on *known* covert errors (historical hacks), this setting serves as a proxy for past events. To truly assess the marginal value of

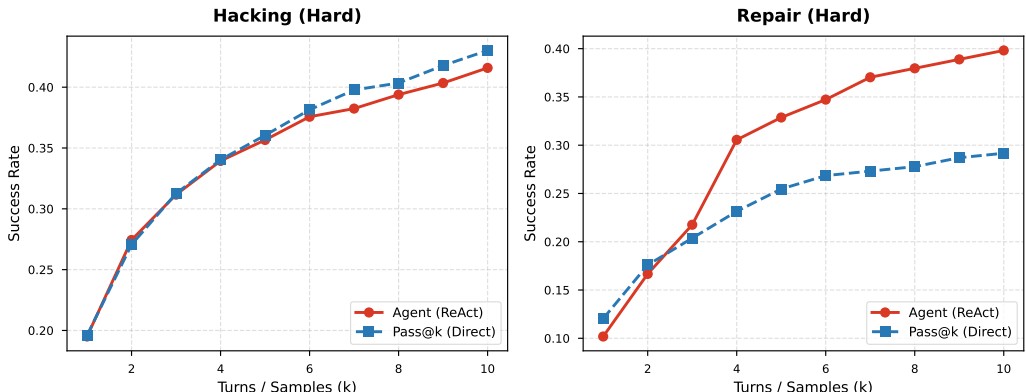

*Figure 6.* Performance Scaling with Increased Computational Budget (Direct Sampling vs. Agentic Interaction). The plots compare the cumulative success rates of GPT-OSS-120B on Hacking (Left) and Repair (Right) tasks over 10 turns.

LLMs in a real-world pedagogical environment, we extend our evaluation to a "Wild" setting: **Zero-Day Hacking**.

In the context of competitive programming education, an "Accepted" (AC) verdict merely signifies that a solution has passed all *standard test cases*. As discussed in the Introduction, this does not guarantee the absence of *covert errors*. A solution containing covert errors typically passes the standard suite but fails on specific, boundary-case inputs.

We constructed the **Zero-Day-Hacking-5K** dataset, containing $5,060$ target submissions. Because zero-day errors are rare under uniform sampling, we bias selection toward problems with a higher likelihood of undiscovered vulnerabilities, based on historical hack frequency, making the evaluation more sensitive to models' abilities. For each problem, we sample historical full-score submissions at a rate of $5\times$ the number of existing hacks in the "Hard" partition of the Hacking dataset. To ensure validity for zero-day discovery, all sampled submissions are re-evaluated against the current UOJ test suite, retaining only those that still achieve full scores. This filtering guarantees that the final targets are robust solutions passing all known test cases. We also ensure that reference solutions are correct on all these new test cases (see Section B.7).

We tasked three representative models: **GPT-OSS-120B**, **Gemini-3-pro-preview**, and **GPT-5**, with performing automated hacking on these solutions. The objective was to generate new test cases capable of exposing potentially *unknown covert errors*. Figure 3 summarizes the models' capacity to discover these previously unknown vulnerabilities. Overall, GPT-OSS-120B is able to uncover errors in over 10% of all submissions.

We further analyze the distribution of GPT-OSS-120B's success rates across individual problems. As illustrated in Figure 5, there is substantial variation in success rates between problems. Notably, GPT-OSS-120B uncovers errors in over 5% of full-score submissions across around 30 prob-

lems, and over 10% of full-score submissions across around 20 problems.

**Implications.** Successful hacks demonstrate that LLMs can move beyond the role of passive solvers to become active *verifiers*. By exposing latent flaws in human-written code, they can help educators to strengthen test cases and reduce misleading false positives. This capability is even more valuable for official contests, where test case quality is closely tied to fairness and outcomes can directly impact the professional careers of students.

# 6. Conclusions

In this work, we introduced UOJ Bench, a challenging benchmark designed to evaluate Large Language Models not merely as problem solvers, but as active components of the competitive programming infrastructure. By distinguishing between overt and covert errors through our Hacking and Repair tasks, we demonstrated that current frontier models possess the reasoning depth necessary to detect and fix deep logical flaws that evade standard test cases, especially when augmented with test-time scaling.

Building on these promising capabilities, a key future direction involves a small-scale deployment of our pipeline directly on online judge systems. Such integration of LLM-based hacking and repair assistants into the live judging loop would provide further validation of their practical utility in enriching test suites and providing pedagogical feedback to human learners.

# Impact Statement

This paper presents work aimed at advancing the evaluation of large language models in competitive programming, with a focus on code hacking and repair using real-world online judge data. The anticipated societal and ethical impacts are consistent with those of prior work in machine learning

benchmarking and educational tooling. We do not foresee significant negative impacts beyond standard considerations. To support transparency and reproducibility, we are willing to provide access to UOJ's internal evaluation API to researchers interested in testing their own models under the same conditions.

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

# A. Prompt Templates

In this section, we provide the specific prompts used for each task in UOJ Bench. Variable contents such as problem descriptions or source code are denoted by brackets (e.g., {PROBLEM_DESCRIPTION}).

## A.1. Task 1: Code Generation Prompts

For the code generation task, we employ a standard competitive programming system prompt designed to encourage efficient and correct algorithms. The `problem` placeholder is substituted with the English problem statement.

---

**Code Generation Prompt**

**Prompt:**
You are an expert C++20 programmer. You will be given a question (problem specification) and will generate a correct C++20 program that matches the specification and gets as many points as possible you can.

### Question:
{problem}

Read the inputs from `stdin`, solve the problem, and write the answer to `stdout` (do not directly test on the sample inputs). Enclose your code within the following delimiters. Ensure that when the C++ program runs, it reads the inputs, runs the algorithm, and writes the output to `stdout`.

```cpp
# YOUR CODE HERE
```

### Answer: (use the provided format with backticks)

---

## A.2. Task 2: Code Hacking Prompts

For the hack task, the model is instructed to find counter-examples. The `problem` and `code` placeholder is substituted with the English problem statement and the target code, respectively.

---

**Code Hacking Prompt**

**Prompt:**
You are an expert at breaking buggy code. You will be given a buggy code and the complete description of the problem it intends to solve. Your job is to find a valid input, in the expected input format and within all input constraints, on which the code fails (either produces a Wrong Answer or runs into Time Limit Exceeded).

Write a python program to print this failing test-case. Enclose your code within delimiters as follows.

```python
# YOUR CODE HERE
```

### Question:
{problem}
### Code:
{code}
### Answer: (use the provided format with backticks)

---

## A.3. Task 3: Code Repair Prompts

For the code repair task, we instruct the model to perform minimal edits to fix the bug. The `problem` and `code` placeholder is substituted with the English problem statement and the target code, respectively.

---

**Code Repair Prompt**

**Prompt:**
You are an expert at fixing bugs in code. You will be given a buggy code and the complete description of the problem it intends to solve. Your job is to modify the code to make it correct while making as few changes as possible. The change must be expressed as a patch file that can be directly applied to the code using the patch command. Do not add any comments or explanations in the patch. Make sure your patch is minimal, i.e., the number of lines of code added or deleted is as small as possible. Enclose your patch within delimiters as follows.

```patch
# YOUR PATCH HERE
```

Here is an example of a patch file. It consists of changes to some example code. It specifies the line numbers of each change, and the removed and added lines.

```patch
@@ -6,6 +6,6 @@
    int sum = 0;

-    for (int i = 0; i <= 5; i++) {
+    for (int i = 0; i < 5; i++) {
        sum += arr[i];
    }
```

### Question:
{problem}

### Code:
{code}

### Answer: (use the provided format with backticks)

---

## A.4. Agentic Evaluation Prompts (ReAct)

In the Agentic setting, the model engages in a multi-turn interaction. We preserve the full conversation history (including previous reasoning and outputs). In each subsequent turn, the environment appends a specific feedback message based on the execution result.

---

**Agentic Hack Feedback Prompt**

The following feedback is appended to the conversation history as a new **User Message**, depending on the execution outcome.

**Case 1: No Python Block Found**
No Python code block found in your response.

Try again! Output a new python code which would generate the correct hack data.

---

**Case 2: Hack Failed (Invalid Input or Target Code Passed)**
The python code generate invalid input or the code can still pass your test. Here is the results

```
{result}
```

Try again! Output a new python code which would generate the correct hack data.

*Note: {result} contains the generated input and the standard output of the target code. Large outputs are truncated.*

---

---

**Agentic Repair Feedback Prompt**

The following feedback is appended to the conversation history as a new **User Message**, depending on the validation outcome.

**Case 1: No Patch Block Found**
No patch block found in your response.

Try again! Output a new patch which would be directly applied to the code given for the first time.

---

**Case 2: Similarity Check Failed**
You made too many changes.

Try again! Output a new patch which would be directly applied to the code given for the first time.

---

**Case 3: Logic Verification Failed (WA/TLE/RE)**
The new code cannot pass all tests. Here is the results

`{result}`

Try again! Output a new patch which would be directly applied to the code given for the first time.

*Note: `{result}` contains the failed test case input, the code's output, and the error verdict (e.g., expected vs. found). Large outputs are truncated.*

---

**Case 4: Patch Application Failed**
Meet error `{e}`

Try again! Output a new patch which would be directly applied to the code given for the first time. *Note: `{e}` contains the error message met when applying the patch.*

---

## B. Data Curation and Evaluation Details

### B.1. Filtering Pipeline

To ensure the Code Hacking (Hard) dataset remains high-quality and compatible with LLM benchmarking standards, we implemented the following filtering criteria:

1. To ensure a rigorous and unified evaluation pipeline, we exclude non-standard formats because they deviate from the widely adopted Single-File, Standard I/O paradigm. These excluded types often require multi-file submissions or non-static hack protocols, which introduce ambiguity and incompatibility with current LLM benchmarking standards.

2. We exclude problems heavily reliant on randomized algorithms (e.g., rolling hashes). Hacks for such problems often rely on brute-forcing collisions against specific modular constants rather than exploiting logical flaws, which does not align with our focus on reasoning.

3. We filter out "self-hacks" (users hacking their own code for recreation) and submissions containing leftover debugging artifacts. To distinguish these from genuine algorithmic errors, we employ a hybrid filtering pipeline combining LLM-based semantic analysis, rule-based pattern matching for known debugging idioms, and targeted manual inspection.

4. Over the years, the UOJ judging infrastructure has undergone significant upgrades, causing some older submissions (with Time Limit Exceeded verdicts) to pass under current hardware conditions. To address this, we re-evaluate every candidate buggy submission against the current judge; any submission that now passes all test cases is discarded.

### B.2. Token Consumption

Table 4 reports the average input and output token consumption per problem for each model across different tasks, together with their corresponding per-million-token pricing. These statistics are used to estimate the average inference cost of each model by combining token usage with provider-listed prices. In particular, we use the token consumption and pricing information in this table to compute the cost axis of the cost–performance plots, enabling a fair comparison of models that differ substantially in both generation length and pricing structure.

*Table 4.* Average input and output token consumption across tasks, along with model pricing. Pricing information is obtained from OpenRouter.

| Model | Generation | | Hacking (Hard) | | Repair (Hard) | | Pricing ($/M tokens) | |
|---|---|---|---|---|---|---|---|---|
| | In (tok) | Out (tok) | In (tok) | Out (tok) | In (tok) | Out (tok) | In | Out |
| GPT-5.2-high | 2.81k | 50.6k | 4.36k | 13.4k | 3.89k | 16.5k | 1.75 | 14.0 |
| Gemini-3-pro-preview | 1.68k | 22.9k | 3.33k | 20.9k | 3.51k | 23.6k | 2.0 | 12.0 |
| GPT-5 | 1.62k | 22.9k | 2.82k | 13.6k | 3.16k | 7.0k | 1.25 | 10.0 |
| GPT-5-high | 1.62k | 35.9k | 2.82k | 20.0k | 3.16k | 13.2k | 1.25 | 10.0 |
| Kimi-k2-thinking | 1.57k | 67.8k | 2.81k | 38.0k | 2.99k | 42.6k | 0.32 | 0.48 |
| GPT-OSS-120B | 1.68k | 10.4k | 2.88k | 5.7k | 3.18k | 2.9k | 0.02 | 0.10 |
| Claude-Opus-4.5 | 1.63k | 1.9k | 2.33k | 1.6k | 2.46k | 1.5k | 5.0 | 25.0 |
| Qwen3-Coder | 1.67k | 3.6k | 2.91k | 0.5k | 3.16k | 0.6k | 0.22 | 0.95 |
| Deepseek-v3.2-thinking | 1.49k | 25.3k | 3.10k | 20.3k | 3.00k | 22.4k | 0.25 | 0.38 |

*Table 5.* Detailed statistics on the number of problems and task instances for each benchmark. Note that the counting protocols are not always directly comparable across benchmarks (e.g., problems, sampled instances, or repair trajectories).

| Benchmark | Generation | Hacking | Repair |
|---|---|---|---|
| LiveCodeBench | 400 | – | – |
| LiveCodeBench Pro | 584 | – | – |
| CodeELO | 398 | – | – |
| REFUTE | – | 324 | – |
| ELABORATION | 8,320 | – | – |
| UOJ Bench | 672 | 1,525 | 716 |

## B.3. Task Instances Statistics

Table 5 provides detailed statistics on the number of problems and task instances included in each benchmark. We note that these quantities are not always directly comparable across benchmarks, since different works adopt different counting protocols (e.g., counting problems, sampled instances, generated repairs, or repair trajectories).

## B.4. Difficulty Assignment

We determine problem difficulty based on the originating source, harmonizing metrics from Luogu, OJ-bench, JOISC, UOJ Native contests, and expert annotations. To ensure consistency across these diverse sources, we employ the following standardization rules:

- **OJ-bench:** We retain the original difficulty annotations for problems sourced from OJ-Bench. Since OJ-Bench does not include an Ultrahard category, we introduce this category by leveraging the internal numeric difficulty ratings from the Luogu platform when available. Specifically, among OJ-Bench problems labeled as Hard, those with a Luogu internal rating of 7 are reclassified as Ultrahard, while the remaining ones are kept as Hard.

- **Luogu:** For problems whose sources can be identified and retrieved on the Luogu platform, we assess difficulty using Luogu's internal numeric rating system published on its official website. These internal levels are mapped to our unified difficulty categories as follows: Easy: 1–3, Medium: 4–5, Hard: 6, Ultrahard: 7.

- **JOISC:** We find no Easy problem in JOISC. Difficulty is inferred from the number of accepted submissions (AC counts), which are collected by crawling the official website: Medium: $\geq 180$, Hard: $65 \sim 180$, Ultrahard: $< 65$).

- **UOJ Contests:** We find no Easy problem in UOJ contests. A problem is labeled Medium if its Acceptance Raio exceeds a problem-specific threshold, which is linearly scaled from $0.3$ to $0.5$ according to the problem ID to account for the growth of the user base over time. A problem is classified as Hard if its number of accepted submissions exceeds another problem-specific threshold, which is scaled from $8$ to $16$ by the same method. All remaining problems are considered Ultrahard.

- **Other Sources:** There are some problems that are not belong to any of the above classes (e.g. other UOJ original problems, some IOI and National Team Training problems). For these problems, their difficulties class is labeled by human experts.

## B.5. LLM-based Bug Classification

In the Data Curation phase for the **Code Repair (Hard)** dataset, we employ an LLM (GPT-5) to classify (Buggy, Fixed) pairs into semantic categories and filter out invalid instances (e.g., hardcoded fixes). The prompt used for this classification is provided below.

---

**Bug Classification Prompt**

**Prompt:**
You are an expert in competitive programming and debugging algorithmic code.

You are a coding assistant. Your task is to classify the difference between two user-submitted c++ codes for the same algorithmic problem. Both codes pass most of the test cases, but one of them fails on carefully designed tests.

You are given:
**Problem Description:**
{problem_desc}

**Code A:**
{code_a}

**Code B:**
{code_b}

Please analyze the small differences between Code A and Code B, and decide the most likely error type that caused Code A to fail while Code B passes.

**Possible error types:**
1. **Hardcode:** the modification is not a genuine bug fix, but simply hardcoding: e.g., deleting debug output, adding ad-hoc checks for specific inputs, or directly returning a fixed value.

2. **Corner Case Error:** Code A only fails on very small, very large, or boundary inputs (e.g., n=0, n=1, maximum values, overflow, precision issues, division by zero).

3. **Careless Error:**

    • Wrong data type (e.g., int vs long long)

    • Wrong variable name (e.g., swapping n and m)

    • Wrong constant value (e.g., INF too small, EPS too large)

    • Forgot to clear or reset arrays/variables

4. **Wrong-but-Works Algorithm:** Code A has a logical or implementation error, but coincidentally works for most inputs.

5. **Randomness-related Error:** Code A uses randomization but fixes a random seed or parameter, so adversarial input can be constructed against it.

6. **Both Wrong:** Neither code is a fully correct solution; both rely on heuristics, pruning, or parameter tuning, and should be excluded. Notice that if the new content in code B has some magic number or constant number, and these number are not for some corner case like n=1, you should select "Both Wrong".

**IMPORTANT:** If you are NOT confident about the category or do not understand the logics of the algorithm, always choose "Both Wrong".

**Your output format must be strict JSON:**
```
{{
"reason":  "<a short explanation of why Code A failed while Code B succeeded>",
"error_type":  "<one of the above types>"
}}
```

---

## B.6. Patch Application

For **Task 3 (Modify)**, the evaluation pipeline involves strict validation steps to ensuring the generated patch is both syntactically valid. But Large Language Models frequently struggle to generate exact line numbers and adhere to strict `diff` formatting standards. To mitigate this, we employ a fault-tolerant patch application pipeline:

1. **Context Inference:** If the generated patch omits line numbers, our pipeline scans the source code to heuristically locate

the matching context block and infers the correct insertion point.

2. **Header Correction:** Since models often hallucinate or miscount the number of added/deleted lines in the hunk header, we ignore the generated counts. Instead, we programmatically recalculate the correct line counts based on the actual content of the patch body.

3. **Fuzzy Application:** We execute the patch using the lenient command:

```
patch --batch --fuzz=5 -l src_path -i patch_path
```

The `--fuzz=5` flag allows for up to 5 lines of context mismatch, and `-l` ignores whitespace differences, maximizing the success rate for semantically correct but syntactically imperfect patches.

### B.7. Zero-Day Validation

To ensure the reference solutions themselves do not fail on the new test cases, newly discovered bugs are strictly verified by the live UOJ judging system rather than relying solely on static analysis.

When a generated hack (a new test case) is successfully submitted, the UOJ infrastructure automatically triggers a global re-evaluation of all historically accepted submissions for that problem. This systemic approach yields two possible outcomes. First, if the platform's official reference solution breaks on the new test case—typically causing the vast majority of previously accepted solutions to fail alongside it—it indicates an invalid test case or a fundamentally flawed problem setup. Such instances are flagged and prompt manual intervention by UOJ administrators. Conversely, if the generated hack isolates and breaks only a few specific submissions (including our target) while the official reference solution successfully passes, the test case is automatically validated by the system as exposing a true covert bug. This rigorous, platform-level verification guarantees that the zero-day vulnerabilities discovered by the LLMs are genuine logical errors in the target code, rather than artifacts of faulty evaluation scripts.

## C. Additional Experiments and Analysis

### C.1. Performance Analysis

Tables 2 and 3 offer critical insights into LLM capabilities across distinct algorithmic reasoning tasks:

**Generation Performance:**  Table 2 clearly shows a steep performance drop for all models as algorithmic problem difficulty increases. While SOTA models excel on **Easy** problems (e.g., Gemini-3-pro-preview at 94.44%), performance plummets dramatically in the **Ultrahard** category (Gemini-3-pro-preview: 18.62%; most others below 10%). This confirms that UOJ Bench's Generation task effectively gauges models' limits in synthesizing complex algorithms.

**Consistent Model Rankings:**  We observe a striking consistency in model rankings across domains. Notably, the performance order in **Generation** (Task 1) is replicated **identically** in **Repair** (Task 3), from the state-of-the-art Gemini-3-pro-preview down to Qwen3-Coder. In **Hacking** (Task 2), the hierarchy remains virtually unchanged, with the only deviation being a minor swap between GPT-5 variants. This "monolithic" performance structure implies that Hacking and Repairing are not orthogonal skills, but rather downstream manifestations of a model's core algorithmic reasoning capacity—stronger coders are inherently better debuggers.

**Overt vs. Covert Errors:**  The consistent performance drop from "Easy" to "Hard" tasks validates the distinction between *overt* and *covert* errors. While frontier models show competence in addressing overt errors (exposed by standard test cases), they struggle significantly with covert errors—subtle bugs that evaded standard testing and required community hacks to uncover. This confirms that detecting and fixing bugs in "ostensibly correct" code poses a much higher reasoning barrier than correcting code that simply fails initial checks.

### C.2. Relationship between Generation and Debugging

To investigate the dependency between constructive and analytical skills, we examine whether the ability to generate a solution is a prerequisite for debugging it.

We calculate conditional success rates for Hack and Repair tasks based on whether the model successfully solved the original problem ($Gen_{\checkmark}$) or failed ($Gen_{\times}$). Table 6 reveals a partial decoupling between generation and debugging capabilities:

*Table 6.* Conditional Success Rates: Hack and Repair Performance based on Generation Outcome

| Model | Overall (%) | Given $Gen_\checkmark$ (%) | Given $Gen_\times$ (%) |
|---|---|---|---|
| **Task 2: Hacking (Hard) Success Rate** | | | |
| Gemini-3-pro-preview | 44.17 | 59.72 | 29.70 |
| GPT-OSS-120B | 20.17 | 33.33 | 17.04 |
| **Task 3: Repair (Hard) Success Rate** | | | |
| Gemini-3-pro-preview | 48.15 | 67.74 | 33.33 |
| GPT-OSS-120B | 9.72 | 10.26 | 9.60 |

**Models can debug problems they cannot solve:** Remarkably, models retain significant debugging potential even when they fail to solve the problem ($Gen_\times$). For instance, Gemini-3-pro-preview achieves a 29.70% Hack rate and 33.33% Repair rate on problems it could not solve itself. This suggests that **verification is distinct from construction**: models can often identify edge cases or fix local logic errors without possessing the capability to synthesize the full algorithm from scratch.

**Models are better at debugging problems they can solve:** While not a strict prerequisite, successfully generating the solution ($Gen_\checkmark$) does amplify debugging performance (e.g., Gemini-3-pro-preview Hacking rises to 59.72%). However, the success rate is far from guaranteed, confirming that finding bugs requires specific testing skills that mere algorithmic understanding does not automatically provide.

**Limitations of Weaker Models:** For weaker models like GPT-OSS-120B, the Repair gap between $Gen_\checkmark$ (10.26%) and $Gen_\times$ (9.6%) is negligible. This implies that even when the model understands the algorithm, it may lack the precise code-editing instructions required to perform a valid fix.

### C.3. Relationship between Hacking and Repairing

We analyze the alignment between bug discovery (Hack) and repair (Repair) in Table 7. The discrepancies between these tasks reveal distinct cognitive modes in LLMs:

*Table 7.* Intersection of Performance on Hack and Repair Tasks. We report the count ($N$) and percentage of problems falling into each category: both tasks successful, only Hacking successful, only Repair successful, or both failed.

| Model | Both Correct (Hack ✓, Repair ✓) | Hack Only (Hack ✓, Repair ✗) | Repair Only (Hack ✗, Repair ✓) | Both Wrong (Hack ✗, Repair ✗) |
|---|---|---|---|---|
| Gemini-3-pro-preview | 62 (28.70%) | 37 (17.13%) | 42 (19.44%) | 75 (34.72%) |
| GPT-OSS-120B | 9 (4.17%) | 40 (18.52%) | 12 (5.56%) | 155 (71.76%) |

**Hacking without Repairing:** The significant portion of "Hack Only" cases (e.g., 17.13% for Gemini-3-pro-preview) confirms that recognizing a failure mode is fundamentally different from fixing it. Models can often identify a violation of problem constraints (finding a counter-example) but lack the precise code-synthesis capability to re-implement the logic correctly.

**Repairing without Hacking:** Surprisingly, models frequently fix bugs without being able to trigger them (19.44% for Gemini-3-pro-preview). This suggests a reliance on pattern-matching heuristics rather than deep semantic understanding. For instance, a model might reflexively apply a standard patch—such as changing `(a-b)%m` to `(a-b+m)%m`—because it recognizes the pattern of a potential error, yet it fails to mathematically construct the specific input (e.g., where `a < b`) required to prove the bug exists.

### C.4. Contamination Analysis

To assess whether the models' performance stems from genuine reasoning capabilities or mere memorization of training data (data contamination), we analyze the success rate relative to the chronological release dates of the problems (or submission dates of the target submissions). A primary concern in evaluating Large Language Models on public benchmarks is the risk that older problems were included in the model's pre-training corpus. If a model relies on memorization, we would expect a

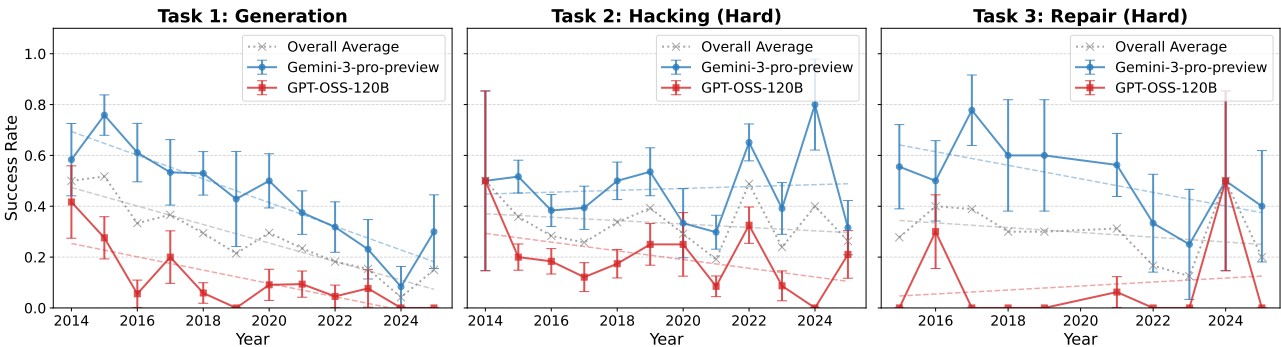

*Figure 7.* Temporal generalization analysis on Hard problems. The plots display the success rates of **Gemini-3-Pro-Preview** and **GPT-OSS-120B** across years. While *Code Generation* shows a performance drop on recent data (suggesting contamination), *Code Hacking* and *Self-Repair* remain stable, indicating reliance on reasoning rather than memorization.

significant performance degradation on problems released after the model's knowledge cutoff date. Conversely, a stable performance trend across time indicates robust generalization. We conducted this analysis using **Gemini-3-pro-preview** and **GPT-OSS-120B**.

An important nuance of the UOJ platform substantially reduces contamination risk for our benchmark. While problem statements and user submissions are publicly accessible, the *problem data*—including hidden judge test cases and Extra Tests used in hacking—are not publicly released. Consequently, even if public repositories or web crawls contain historical submissions, they would still lack access to the exact hidden inputs required for successful hacking. Similarly, for the Repair task, publicly available submissions do not naturally provide aligned *(buggy submission, corrected fix)* pairs. Constructing such supervision requires nontrivial reconstruction from submission histories rather than direct memorization from raw web data. Therefore, contamination risk for Hacking and Repair is inherently much lower than for standard code generation. Our empirical findings in Figure 7 further support this observation.

To ensure a fair comparison and mitigate the confounding variable of problem complexity fluctuating over time (e.g., problems naturally becoming more complex in recent years), we restricted our dataset exclusively to problems and submissions classified as **Hard**. We partition these test samples by year based on their release dates or submission timestamps and calculate the pass rate for each temporal bin. Figure 7 illustrates the temporal performance trends across the three tasks. The results reveal a divergence in behavior:

1. **Code Generation:** We observe a discernible decline in success rates for more recent problems. This suggests that the models' high performance on older "Hard" problems is partially attributable to memorization of canonical solutions present in the training data.

2. **Code Hacking and Self-Repair:** In contrast, these tasks exhibit a stable performance trend across the timeline. This stability indicates that Hacking and Repair rely less on retrieving memorized patterns and more on active reasoning, logic verification, and understanding specific failure modes. These capabilities demonstrate robust generalization to unseen, zero-day scenarios.

### C.5. Impact and Robustness of Edit Distance Constraint

To ensure our 0.95 similarity threshold (based on Levenshtein distance) does not artificially narrow the dataset to superficial, typo-level fixes, we first evaluated the sensitivity of this design choice. We established this threshold primarily to filter out complete code refactors, as rewriting a solution from scratch bypasses the targeted debugging and patching objectives of the repair task. Manual inspection confirms that within the 95% constraint, users successfully implement crucial algorithmic corrections, handle complex edge cases, and resolve substantial logic flaws. Furthermore, our data exhibits a naturally polarized distribution: among submission pairs with $> 90\%$ similarity, 85% exceed 95%, and 65% exceed a strict 98% similarity. To test robustness, we evaluated Gemini-3-Pro-Preview under a tighter 0.98 threshold. Under this strict constraint, the model's repair accuracy was 55.6%, highly consistent with the 53.6% achieved under the default 0.95 threshold. This demonstrates that the benchmark's difficulty is robust and not overly sensitive to the exact threshold value.

Furthermore, we analyze the repair failures of Gemini-3-pro-preview on similarity violations to investigate whether the edit distance constraint disproportionately affects specific error types, particularly Time Limit Exceeded (TLE). In the Easy track, the ratio of similarity violations for TLE compared to other errors is 10.2% (10/98) versus 13.2% (53/402). In the Hard track, these ratios are 10.9% (5/46) versus 4.7% (8/170). These figures indicate that the rate of similarity violations remains relatively consistent across error categories, confirming that the edit budget is not the primary bottleneck for resolving TLEs. Instead, the true difficulty of TLE cases stems from the fundamental nature of the bugs. TLEs in our dataset typically arise from inefficient algorithms where time complexity severely degrades under specific adversarial edge cases. Resolving these performance bottlenecks requires a holistic rethinking of the algorithm's complexity, whereas Wrong Answer (WA) or Runtime Error (RE) cases typically involve localized logic flaws that are inherently easier for LLMs to diagnose and patch.

## C.6. Patch vs. Full Code Generation

To validate our choice of using patch as the repair method, we evaluate two output modes for Task 3: **Full Code Rewrite** (regenerating the entire file) versus **Patch Generation** (outputting minimal `diffs`).

*Table 8.* Performance Comparison: Full Rewrite vs. Patch Generation

| Model | Full Rewrite (Pass@1) | Patch Generation (Pass@1) | Improvement |
|---|---|---|---|
| Gemini-3-pro-preview | 7.87 | 48.15 | **6.1**× |
| GPT-OSS-120B | 0.46 | 9.72 | **20.1**× |

As illustrated in Table 8, the results overwhelmingly favor **Patch Generation**, with success rates increasing by an order of magnitude compared to Full Code Rewrite. GPT-5 improves from 3.57% to 32.14%, while the open-source model sees an even larger relative gain.

These results validate our decision to adopt Patch Generation as the standard metric for the Repair task, as it better isolates the model's debugging capability from its code completion stability.

## C.7. Language Impact: English vs. Chinese

To verify translation quality, we manually inspected a diverse sample of the translated problem statements, including the most complex ones. We found that the LLM translations were excellent—preserving all critical mathematical constraints, subtle hints, and thematic background stories.

Furthermore, to assess whether translating the original Chinese problem statements affects reasoning, we compare model performance on three tasks using the original Chinese prompts and statements versus the translated English versions.

*Table 9.* Impact of Prompt Language across Tasks (Pass@1)

| Model | English Prompt | Chinese Prompt | $\Delta$ (CN - EN) |
|---|---|---|---|
| ***Task 1: Generation*** | | | |
| Gemini-3-pro-preview | 38.84 | 40.18 | +1.34 |
| GPT-OSS-120B | 12.35 | 13.99 | +1.67 |
| ***Task 2: Hacking (Hard)*** | | | |
| Gemini-3-pro-preview | 44.17 | 40.92 | -3.25 |
| GPT-OSS-120B | 20.17 | 19.89 | -0.28 |
| ***Task 3: Repair (Hard)*** | | | |
| Gemini-3-pro-preview | 48.15 | 44.91 | -3.24 |
| GPT-OSS-120B | 9.72 | 8.80 | -0.92 |

Table 9 presents a nuanced landscape of language impact. For Task 1 (Generation), we observe a slight performance advantage in the original Chinese prompts ($\Delta \approx +1.5\%$). This is expected, as the source problems originate from a Chinese platform, and the native text may retain subtle semantic nuances.

*Table 10.* Success Rate (%) across Different Error Types. Breakdowns show performance when the target submission has a specific verdict: Wrong Answer (WA), Time Limit Exceeded (TLE) or Runtime Error (RE).

| Model | Task 2: Hacking (Hard) | | | Task 3: Repair (Hard) | |
|---|---|---|---|---|---|
| | WA | TLE | RE | WA | TLE |
| GPT-5.2-high | 66.31 | 42.47 | 64.06 | 55.88 | 39.13 |
| Gemini-3-pro-preview | 47.54 | 35.24 | 56.25 | 48.24 | 47.83 |
| GPT-5-high | 41.85 | 25.30 | 46.88 | 41.18 | 30.43 |
| GPT-5 | 42.00 | 28.92 | 46.88 | 36.47 | 19.57 |
| Kimi-k2-thinking | 32.46 | 18.07 | 37.50 | 30.59 | 13.04 |
| Deepseek-v3.2-thinking | 30.46 | 13.86 | 34.38 | 26.47 | 15.22 |
| GPT-OSS-120B | 22.46 | 12.95 | 34.38 | 8.82 | 13.04 |
| Claude-Opus-4-5 | 11.23 | 3.01 | 17.19 | 8.24 | 2.17 |
| Qwen3-Coder | 6.46 | 2.41 | 14.06 | 4.71 | 0.00 |

For the more complex reasoning tasks of Hacking and Repair, we observe a slight but consistent advantage for English prompts across both models. In particular, performance under English prompts is marginally higher, with differences on the order of 1–3 percentage points. A plausible explanation is that English prompts may align more closely with the training distribution of contemporary large language models, leading to more stable reasoning and fewer misunderstandings in edge-case–heavy tasks. Nevertheless, the magnitude of these differences remains small, and no substantial performance gap emerges across tasks or models.

## C.8. Test-Time Scaling

Figure 8 presents the performance of Gemini-3-pro-preview and GPT-OSS-120B under test-time scaling, where the horizontal axis denotes the number of samples $k$ and the vertical axis reports cumulative success rates. The curves illustrate how performance improves as additional samples are allocated at inference time, for both hacking and repair tasks.

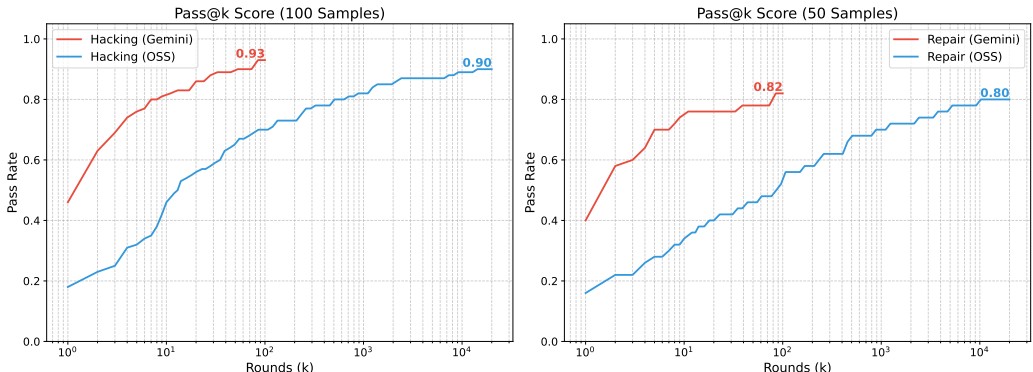

*Figure 8.* Pass@$k$ performance of Gemini-3-pro-preview and GPT-OSS-120B. The figure illustrates the test-time scaling capability of the selected models on the hard subsets of the hacking and repair datasets.

## C.9. Impact of Difficulties, Error Types and Problem Types

To pinpoint the specific bottlenecks in automated debugging, we dissect the performance across three dimensions: error types, problem types, and problem difficulty.

**Error Types (WA vs. TLE/RE):** Table 10 breaks down performance by the verdict of the buggy submission. We observe that Time Limit Exceeded (TLE) errors are significantly more difficult to both hack and repair than Runtime Errors (RE) or Wrong Answers (WA). This is consistent with the intuition that addressing TLE typically requires substantive algorithmic optimization or a rethinking of the overall approach, whereas hacking or repairing RE and WA often involves identifying and exploiting localized issues such as boundary conditions or logic flaws. Models show a higher success rate in triggering Runtime Errors, as these often result from unhandled edge cases (e.g., division by zero, array out of bounds) that are easier to target with fuzzing-like strategies than the precise logic required to prove a Wrong Answer.

*Table 11.* Performance Analysis across Problem Characteristics and Difficulty Levels (Success Rate %). We compare standard input/output problems versus those requiring special checkers, and breakdown performance across four difficulty tiers.

| Model | Task | Grading Type | | Problem Difficulty | | | |
|---|---|---|---|---|---|---|---|
| | | Standard | Special | Easy | Medium | Hard | Ultra Hard |
| GPT-5.2-high | | | | 88.89 | 76.84 | 51.33 | 27.33 |
| Gemini-3-pro-preview | | 40.43 | 30.97 | 94.44 | 80.00 | 46.90 | 18.62 |
| GPT-5-high | | 26.30 | 22.12 | 88.89 | 64.21 | 28.76 | 9.01 |
| GPT-5 | | 24.69 | 20.35 | 77.78 | 63.16 | 25.66 | 8.71 |
| Kimi-k2-thinking | Generation | 22.18 | 13.27 | 88.89 | 55.79 | 23.89 | 4.80 |
| Deepseek-v3.2-thinking | | 19.50 | 15.04 | 77.78 | 58.95 | 18.14 | 4.50 |
| GPT-OSS-120B | | 13.42 | 7.08 | 77.78 | 40.00 | 11.50 | 1.50 |
| Claude-Opus-4-5 | | 6.80 | 6.19 | 50.00 | 23.16 | 5.31 | 0.60 |
| Qwen3-Coder | | 2.68 | 0.88 | 33.33 | 9.47 | 0.44 | 0.00 |
| GPT-5.2-high | | 59.28 | 53.60 | 87.88 | 75.71 | 56.08 | 51.03 |
| Gemini-3-pro-preview | | 46.36 | 28.00 | 87.88 | 63.27 | 45.24 | 31.44 |
| GPT-5-high | | 38.87 | 22.40 | 78.79 | 57.14 | 35.45 | 25.97 |
| GPT-5 | | 39.31 | 29.60 | 75.76 | 59.18 | 34.92 | 28.70 |
| Kimi-k2-thinking | Hacking | 29.42 | 19.20 | 69.70 | 39.29 | 26.98 | 21.18 |
| Deepseek-v3.2-thinking | | 26.71 | 16.00 | 69.70 | 47.96 | 23.28 | 13.90 |
| GPT-OSS-120B | | 20.63 | 16.80 | 57.58 | 35.71 | 18.52 | 11.85 |
| Claude-Opus-4-5 | | 9.88 | 2.40 | 45.45 | 14.80 | 7.41 | 5.01 |
| Qwen3-Coder | | 5.86 | 4.00 | 24.24 | 7.65 | 4.76 | 4.10 |
| GPT-5.2-high | | 51.09 | 59.38 | 50.00 | 61.70 | 45.07 | 53.26 |
| Gemini-3-pro-preview | | 50.00 | 37.50 | 83.33 | 70.21 | 53.52 | 30.43 |
| GPT-5-high | | 39.13 | 37.50 | 83.33 | 53.19 | 39.44 | 28.26 |
| GPT-5 | | 30.98 | 43.75 | 33.33 | 48.94 | 30.99 | 26.09 |
| Kimi-k2-thinking | Repair | 25.54 | 34.38 | 50.00 | 44.68 | 22.54 | 19.57 |
| Deepseek-v3.2-thinking | | 24.46 | 21.88 | 33.33 | 34.04 | 23.94 | 18.48 |
| GPT-OSS-120B | | 8.70 | 15.63 | 16.67 | 10.64 | 7.04 | 10.87 |
| Claude-Opus-4-5 | | 7.61 | 3.13 | 16.67 | 12.77 | 8.45 | 2.17 |
| Qwen3-Coder | | 3.80 | 3.13 | 16.67 | 2.13 | 2.82 | 4.35 |

**Problem Types (Standard vs. Custom Checker):**   Table 11 compares standard input/output problems against those requiring custom checkers. Models typically perform better on Standard problems in the hacking task. In custom checker scenarios—where multiple valid outputs may exist for a single input—the hacking task becomes more challenging, as the model must construct an input for which all valid outputs differ from the buggy program's output, or cause the program to emit an invalid output format. In contrast, the problem grading type has little impact on performance in the repair task. This is because repair primarily requires correcting the underlying algorithmic or implementation flaws in the code itself, which is largely independent of whether the final output is checked against a single canonical answer or a set of valid alternatives.

**Impact of Problem Difficulty:**   As shown in Table 11, there is a clear correlation between the original problem's difficulty and the success of debugging tasks. While Hacking and Repair performance degrades as problem difficulty increases from Easy to Ultra Hard, the drop is less precipitous than in the Generation task (Task 1). This suggests that even for extremely complex problems (Ultra Hard) where models fail to generate a solution from scratch, they retain some capacity to analyze and interact with existing code.

### C.10. Educational Validation and Real-World Interaction

While our primary evaluation establishes the foundational capabilities of LLMs to find and fix covert bugs, we also seek to demonstrate the downstream pedagogical utility of this automated pipeline.

To provide concrete empirical evidence for the educational value of the platform's hacking mechanism, we analyzed historical user behavior on the UOJ platform. Querying the database revealed a total of $2,193$ successful human-generated hacks. Among these, in 786 cases (35.8%), the original author produced a subsequent submission that successfully achieved a full score against the newly hardened test suite (which integrates the adversarial hack). Whether the user fixed the code prompted directly by the hack notification, or was actively refining their approach independently, this data illustrates that exposing latent bugs via concrete counterexamples is a core component of the platform's continuous learning and debugging loop, actively driving students toward truly robust solutions.

Furthermore, our Zero-Day Hacking experiments (Section 5) demonstrate that LLMs can now successfully automate this exact process. As a preliminary real-world validation, we deployed a subset of our LLM-generated zero-day hacks directly to the live UOJ platform. In practice, the UOJ system automatically messages users when their submissions are successfully hacked. Upon deployment, some affected students received these automated notifications, read the LLM-generated counterexamples, and successfully patched the latent bugs in their original code. This real-world interaction confirms that by automating the generation of valid adversarial test cases, LLMs can effectively scale an educational feedback mechanism that is already empirically validated by historical platform data.

### C.11. Impact of Zero-Day Hacking Sampling Method

Since we bias selection toward problems with a higher likelihood of undiscovered vulnerabilities, we conducted an additional evaluation on a uniformly random sampled subset of 4,854 accepted submissions. On this unbiased set, GPT-OSS-120B achieved a Pass@10 success rate of 4.12% (200/4854). For reference, the rate reported in our paper using the biased Zero-Day-Hacking-5K subset was 5.32% (269/5060). While slightly lower under uniform sampling, the success rate remains highly significant. Our original non-uniform sampling was intended to improve evaluation efficiency by filtering out "low-signal" problems (e.g., pure combinatorics). We will include this uniform baseline in the final version.

## D. Case Study

### D.1. Submissions that are Hard to Hack

We manually examine cases that Gemini-3-pro-preview fails to hack after 100 independent attempts. One representative example is discussed below.

We observe that one such submission corresponds to hack ID `11445`[2] for problem `138`[3] on UOJ. This case exemplifies a class of implementations that are theoretically incorrect but practically difficult to exploit.

The problem permits substantial preprocessing on a graph, during which many edges may be contracted and represented using aggregated weights. All computations are performed over the finite field $\mathbb{F}_{998244353}$. The hacked solution adopts an aggressive contraction strategy: multiple original edges are merged into a single weighted edge that stores two accumulated values, whose ratio is later used as the effective edge weight. This approach implicitly assumes that the denominator is always nonzero in $\mathbb{F}_{998244353}$.

However, the implementation fails to handle the case where the denominator becomes zero modulo 998244353, rendering the division undefined. Importantly, this flaw is not triggered by simple or local input patterns. The denominator is formed by combining contributions from many edges during preprocessing and becomes zero only when a carefully constructed input causes the accumulated coefficient to vanish modulo 998244353. Satisfying this condition requires a strong algebraic constraint on the input and is highly sensitive to the specific contraction structure.

Consequently, the set of inputs that expose this bug is extremely sparse, making the corresponding hack data difficult to construct without detailed knowledge of the solution's internal invariants. This example highlights the limitation of brute-force test-time scaling when faced with deeply buried, structure-dependent vulnerabilities.

---

[2]https://uoj.ac/hack/11445
[3]https://uoj.ac/problem/138

