# OpenReview forum: "Beyond Problem Solving: UOJ-Bench for Evaluating Code Generation, Hacking, and Repair in Competitive Programming"
_ICML.cc/2026/Conference — ICML 2026 regular_

### Official Review · Reviewer_B9QX · 2026-03-09

**Soundness:** 3
**Presentation:** 3
**Significance:** 2
**Originality:** 3
**Overall Recommendation:** 4
**Confidence:** 3

**Summary:**

This paper introduces UOJ-Bench, a benchmark built from real UOJ user submissions to evaluate LLMs on three competitive-programming tasks: code generation, code hacking (generating counterexample inputs that break buggy code), and code repair (producing minimal patches to fix buggy code). A central design idea is the distinction between overt errors, which are caught by standard test cases, and covert errors, which pass standard tests but can be exposed through UOJ’s community hack mechanism. The benchmark contains 672 generation problems, 479 Easy and 1,046 Hard hacking instances, and 500 Easy and 216 Hard repair instances. The paper also introduces Zero-Day-Hacking-5K, a set of 5,060 accepted/full-score submissions used to test whether LLMs can discover previously unknown latent bugs. Across a range of frontier models, the paper reports that single-shot performance remains below 60% on Easy hacking/repair and below 50% on Hard instances, while test-time scaling can push success rates above 90% at high inference cost. In the zero-day setting, GPT-OSS-120B under test-time scaling uncovers errors in over 10% of sampled submissions and over 5% of full-score submissions across roughly 30 problems, which the authors interpret as evidence that LLMs may provide complementary bug-finding signals beyond standard online-judge test suites.

**Compliance With Llm Reviewing Policy:**

Affirmed.

**Final Justification:**

Author response resolved my concerns.

**Key Questions For Authors:**

1. **Sensitivity of the similarity threshold**: Have you analyzed how the composition and size of the Hard repair dataset change under alternative similarity thresholds (e.g., 0.90 or 0.98)? Understanding this sensitivity would clarify how much the repair task's difficulty and scope depend on this design choice, and a robustness check would strengthen confidence in the benchmark.
2. **Validation of LLM-based filtering**: What is the accuracy of the LLM-assisted classification and filtering pipeline used to label repair pairs (Section 3.2)? Even a small-scale manual audit with agreement statistics would substantially increase confidence in data quality. If such validation was performed but not reported, please share the results.
3. **Zero-day hacking base rates**: Can you provide the zero-day hacking success rate under uniform random sampling of accepted submissions (rather than the biased sampling strategy used for Zero-Day-Hacking-5K)? This would help calibrate the practical impact of LLM-based hacking on a typical online judge workload, and would clarify how much of the reported 10%+ success rate is attributable to the sampling design.
4. **Educational validation**: Since the paper motivates hacking and repair as tools for learner support, do you have any evidence, even small-scale, that model-generated hacks or repairs actually improve student debugging or learning outcomes?

**Limitations:**

Yes. The paper discusses limitations and societal impact at a reasonable level for this type of benchmarking work. While the discussion could be more specific about scope, generalizability, and possible misuse of hacking-oriented evaluations, I do not view these as major concerns.

**Strengths And Weaknesses:**

Strengths:
1. The benchmark construction is technically solid and realistic. It is built from real UOJ user submissions and real successful hacks rather than synthetic buggy code, and evaluation is performed through UOJ’s native judging pipeline with strict time limits and support for custom checkers. This gives the benchmark stronger ecological validity than many prior competitive-programming evaluations.
2. The paper has an originality component in its evaluation perspective. The main novelty is a new benchmark formulation that asks whether models can find hidden bugs and minimally repair buggy code. The zero-day hacking setting is also an interesting extension because it tests whether models can uncover previously unknown errors in accepted submissions.
3. The paper is generally well written and easy to follow. The motivation, benchmark construction, and empirical setup are presented clearly, and the overall narrative is coherent. In particular, the cost-performance tradeoff and the distinction between ordinary judging and bug-finding are communicated well.

Weaknesses:
1. The practical significance beyond competitive programming is limited. The benchmark operates on isolated competitive-programming submissions, which are essentially single-file problem-solving artifacts rather than realistic multi-file software projects. As a result, the claims about broader real-world software-engineering utility should be interpreted cautiously. Moreover, the empirical findings are somewhat descriptive rather than especially deep. The main conclusions are that current models are still weak on hacking and repair, that test-time scaling helps substantially, and that zero-day hacking can sometimes uncover missed bugs. These are useful observations, but they are not very surprising findings or implications.
2. Some benchmark construction choices would benefit from stronger validation. For the Hard repair set, the paper matches buggy submissions to later successful submissions by the same user using a similarity threshold of 1 - d_edit / L >= 0.95, but it does not report any sensitivity analysis for alternative thresholds. As a result, it is unclear how much the final repair distribution depends on this design choice. The paper also states that the repair pairs are jointly classified and filtered with LLM assistance, but it does not report manual validation statistics or agreement measures for this labeling step. In addition, the Easy repair set is randomly sampled from a filtered pool of about 10,000 candidates without analysis of how representative that sample is across problem types or difficulty levels.
3. The paper’s educational motivation is compelling, but the educational claims remain largely motivational rather than empirically validated. The introduction and conclusion frame hacking and repair as tools that could support learners, improve feedback, and democratize programming education, yet the paper does not include a user study, classroom study, or direct pedagogical evaluation. As a result, the work convincingly establishes benchmark and model evaluation value, but not actual educational impact.

---

> ### Author Rebuttal · Authors · 2026-03-31
>
> ###
>
> We thank the reviewer for acknowledging the strong ecological validity, realism, and original perspective of our benchmark. We address your concerns below, providing new empirical evidence that we believe thoroughly strengthens our claims.
>
> **W1: Practical significance beyond CP is limited (single-file problem solving).**
> We agree that competitive programming focuses on single-file problem solving, which differs from multi-file software engineering. However, the core cognitive skills evaluated here, such as designing algorithms, identifying edge cases, and understanding time complexity bottlenecks, are also fundamental to software engineering.
>
> Furthermore, we argue that having practical significance within CP itself is already meaningful. CP is a widely adopted training ground that cultivates core problem-solving abilities in programmers, which are directly relevant to real-world software engineering practice. Our benchmark serves as a testbed to evaluate whether current LLMs can effectively support this important and widely practiced activity.
>
> **W2 & Q1: Sensitivity of the similarity threshold (0.95).**
> Thank you for this insightful suggestion. We conducted a robustness check by increasing the similarity threshold to a strict 0.98. Under this tighter threshold, the repair accuracy of Gemini-3-Pro-Preview was **55.6%**, which is highly consistent with the **53.6%** reported in the paper (using 0.95). This indicates the benchmark’s difficulty is not overly sensitive to this design choice. Our data shows a polarized distribution: among pairs with >90% similarity, 85% exceed 95%, and 65% exceed 98%. We set the threshold at 0.95 simply to filter out cases where users completely rewrote their code from scratch, as complete refactors do not test targeted debugging capabilities.
>
> **W2 & Q2: Validation of LLM-based filtering.**
> To validate our LLM-assisted filtering pipeline, we conducted a manual audit of the classification results. We found that the LLM distinguishes "Both Wrong", "Wrong-but-Works", “Hardcode” and “Randomness-related Error” cases with high precision. However, for the "Careless" and "Corner Case" categories, the error rate is relatively higher due to their inherently subtle semantic differences, which make them difficult to differentiate even for advanced models. Fortunately, these errors do not affect the composition of the dataset. We are currently manually verifying a larger set of samples and will report the complete validation metrics in Section 3.2 of the final version.
>
> **W2 & Q3: Zero-day hacking base rates under uniform sampling.**
> To calibrate the impact of our sampling design, we conducted an additional evaluation on a uniformly random sampled subset of 4,854 accepted submissions. On this unbiased set, GPT-OSS-120B achieved a Pass@10 success rate of **4.12%** (200/4854). For reference, the rate reported in our paper using the biased Zero-Day-Hacking-5K subset was **5.32%** (269/5060). While slightly lower under uniform sampling, the success rate remains highly significant. Our original non-uniform sampling was intended to improve evaluation efficiency by filtering out "low-signal" problems (e.g., some combinatorics problems have very low hack success rate). We will include this uniform baseline in the final version.
>
> **W3 & Q4: Educational validation and learner outcomes.**
> We argue that our current claims establish the *foundational model capabilities* required for educational tools, rather than serving as a direct pedagogical user study. Before evaluating real-world educational interventions, it is first necessary to establish if models possess the fundamental capability to perform these tasks (finding and fixing covert bugs).
>
> Regarding educational value, the hack mechanism has been deployed on UOJ for many years, and our benchmark just evaluates whether LLMs can accelerate the human process of finding bugs. To provide concrete empirical evidence for the educational value of the hack mechanism, we conducted an analysis of historical user behavior on the UOJ platform. By querying the database, we found a total of 2,193 successful human-generated hacks. Among these, in **786 cases (35.8%)**, the original author produced a subsequent submission that successfully achieved a full score (100) against the newly hardened test suite (which includes the hack).
>
> Our benchmark shows that LLMs can now automate this exact process. In fact, after submitting our paper in January, we submitted several LLM-generated zero-day hacks on the platform. All the affected users received automated notifications, and in many cases, they examined the generated counterexamples and successfully fixed the bugs in their code. While we do not yet have quantitative statistics, these observations provide preliminary evidence that LLMs can meaningfully support this feedback loop in practice.

---

> > ### Author Rebuttal · Reviewer_B9QX · 2026-04-03
> >
> > I thank the authors for their thorough and well-supported rebuttal. The new experiments and analyses effectively address my key concerns. I am happy to raise my score accordingly.

---

> > > ### Author Response · Authors · 2026-04-08
> > >
> > > Thank you very much for your time, your constructive feedback throughout the review process, and for raising your score. We are thrilled that the new experiments and analyses effectively addressed your concerns. We will make sure all of these additions and clarifications are carefully integrated into the final version of the manuscript.

---

### Official Review · Reviewer_rinC · 2026-03-10

**Soundness:** 4
**Presentation:** 3
**Significance:** 3
**Originality:** 3
**Overall Recommendation:** 4
**Confidence:** 3

**Summary:**

This paper introduces UOJ-Bench, a benchmark designed to evaluate LLMs  across three complementary competitive-programming tasks: code generation, code hacking, and code repair. The benchmark is constructed using real-world submissions from the UOJ platform and leverages its native judging infrastructure for evaluation. UOJ-Bench emphasizes debugging and adversarial reasoning, including the detection of covert errors that pass standard test suites but fail under adversarial inputs.
The authors evaluate eight frontier LLMs, analyzing the trade-offs between direct inference, agentic workflows, and test-time scaling.
The work successfully shifts the evaluation paradigm from "model-as-solver" to "model-as-educational-assistant," offering a rigorous assessment of LLMs' ability to provide value beyond traditional judging infrastructure.

**Compliance With Llm Reviewing Policy:**

Affirmed.

**Final Justification:**

The author's response addressed my concern. I will keep the score.

**Key Questions For Authors:**

How representative is UOJ-Bench, compared with other major CP ecosystems like Codeforces or AtCoder?

Since the motivation is partially educational, do the authors plan to evaluate how human students actually interact with these LLM-generated hacks?

How do you plan to handle the inevitable benchmark maintenance as new problems are added to UOJ and models are updated?

**Limitations:**

Yes

**Strengths And Weaknesses:**

Strength

1. This paper presents a well-motivated benchmark construction pipeline. The dataset is derived from real competitive programming submissions and includes authentic debugging scenarios, which increases validity compared with synthetic benchmarks.
2. The distinction between overt and covert errors is a great contribution. The "Zero-Day Hacking" task is interesting, demonstrating that LLMs can uncover latent bugs in historically "perfect" solutions, thereby strengthening existing test suites.
3. The data curation process is notably rigorous, including a 0.95 similarity constraint for repair pairs to ensure they are minimal patches rather than rewrites.
4. The paper is well-structured. The cost-performance analysis (Figure 2) and the detailed comparison with existing benchmarks (Table 1) provide immediate, actionable insights into the current state of the field.
5. This paper includes specific prompt templates, filtering pipelines, and a temporal contamination analysis (Figure 7), which make it more reproducible.

Weakness

1. Although the paper reports performance metrics across tasks and difficulty levels, the analysis of why models fail is relatively limited.  A deeper qualitative investigation into why models fail in hacking or repair tasks would further illuminate the specific reasoning bottlenecks of current models.
2. The benchmark is currently fpcues on C++ language. While significant, the minimal patch paradigm and specific "hack" patterns might not generalize perfectly to other programming languages (e.g., Python/Java) or broader project-level contexts where bugs can span multiple files.
3. Due to inference costs, test-time scaling and agentic evaluations were restricted to smaller subsets (100 and 50 samples, respectively). This limits the statistical confidence when comparing these advanced workflows against the broader dataset.
4. The cost disparity between LLM-based judging (100k/year) and traditional CPU judging (500/year) is massive. The cost may limit the benchmark's practical application.

---

> ### Author Rebuttal · Authors · 2026-03-31
>
> ###
>
> We deeply appreciate your recognition of the benchmark's rigorous data curation, the novel distinction between overt/covert errors, and the immediate insights provided by our cost-performance analysis.
>
> **W1: Limited qualitative analysis of why models fail.**
> We appreciate this constructive suggestion. In the next revision, we will expand Appendix D (Case Study) to include more examples and add a qualitative analysis of the failure modes. This may provide deeper insights into the specific reasoning bottlenecks of current LLMs.
>
> **W2: Focus on C++ and generalization to other languages.**
> We acknowledge this limitation. Competitive programming heavily biases toward C++ due to strict time limits. As a comparison, in the REFUTE benchmark, 317 of 324 samples are in C++. While our findings are rooted in C++, the logical reasoning required to identify boundary conditions and algorithmic flaws is largely language-agnostic. We will explicitly note the focus on single-file C++ as a limitation for generalizing to multi-file projects in Python/Java.
>
> **W4: Cost disparity limits practical application.**
> We agree that a $100k/year LLM judging cost is currently prohibitive compared to traditional CPU judging. However, API costs are dropping rapidly. Our benchmark is designed to establish the foundational *capability* of LLMs to act as active judges and debuggers, serving as a forward-looking evaluation for when inference becomes economically feasible.
>
> **Q1: Representativeness compared with Codeforces or AtCoder?**
> UOJ-Bench is highly representative of advanced competitive programming, containing many problems at the level of the IOI. Some problems directly come from past IOI contests. Many other problems are from major Chinese competitions (used for selecting members of the Chinese IOI team).
>
> Regarding the hack mechanism, while Codeforces supports hacking only within a short window during live contests, UOJ maintains hacking as an open, ongoing mechanism that allows continuous participation. As a result, it may accumulate a richer repository of covert bugs that may take a long time to uncover, which is well aligned with the goals of our benchmark.
>
> **Q2: Human interaction with LLM-generated hacks?**
>
> To provide concrete empirical evidence for the educational value of the hack mechanism itself, we conducted an analysis of historical user behavior on the UOJ platform. By querying the database, we found a total of 2,193 successful human-generated hacks. Among these, in **786 cases (35.8%)**, the original author produced a subsequent submission that successfully achieved a full score (100) against the newly hardened test suite (which includes the hack). Whether the user fixed the code prompted directly by the hack notification, or was actively refining their approach, this data illustrates that exposing latent bugs is a core component of the platform's continuous learning and debugging loop, driving students toward truly robust solutions.
>
> Our benchmark shows that LLMs can now automate this exact process. In fact, after submitting our paper in January, we submitted several LLM-generated zero-day hacks on the platform. All the affected users received automated notifications, and in many cases, they examined the generated counterexamples and successfully fixed the bugs in their code. While we do not yet have quantitative statistics, these observations provide preliminary evidence that LLMs can meaningfully support this feedback loop in practice.
>
> **Q3: Benchmark maintenance plan.**
> Because our data curation pipeline is highly automated, we plan to release periodic updates to UOJ-Bench on Code Generation and Zero-Day-Hacking. We will harvest new problems and user submissions to ensure the benchmark remains fresh and uncontaminated as model capabilities grow.

---

> > ### Author Rebuttal · Reviewer_rinC · 2026-04-04
> >
> > Thanks for the response. It would be great to include the above in the final version of the paper. I will maintain my score.

---

> > > ### Author Response · Authors · 2026-04-08
> > >
> > > Thank you for your continued support and for reviewing our rebuttal. We deeply appreciate your valuable feedback, which has helped strengthen our work. We fully confirm that all the additional statistics, clarifications, and new analyses discussed in the rebuttal will be explicitly included in the final version of the paper.

---

### Official Review · Reviewer_z9ps · 2026-03-13

**Soundness:** 3
**Presentation:** 3
**Significance:** 4
**Originality:** 4
**Overall Recommendation:** 5
**Confidence:** 4

**Summary:**

The authors focus on a pressing question regarding the utility of LLMs in competitive programming beyond standard code generation. Overall, the authors investigate the concept of evaluating models on code hacking and repair using a novel benchmark, UOJ-Bench, derived from real-world submissions. The paper provides valuable insights into the limitations of frontier models in detecting covert errors and highlights the economic bottlenecks of test-time scaling.

**Compliance With Llm Reviewing Policy:**

Affirmed.

**Key Questions For Authors:**

1. Translation Fidelity: In Section 3.2, you mention using an LLM to translate problem statements from Chinese to English. Given the highly specific mathematical vocabulary in competitive programming, how did you verify that no critical constraints or subtle hints were lost or altered during translation?
2. Zero-Day Verification: For the "Zero-Day Hacking" task mentioned in the Introduction, how are the newly discovered bugs verified? Are they manually audited by human experts, or purely validated by the UOJ system, which might itself possess faulty reference solutions for those specific edge cases?
3. Data Contamination Nuance: Appendix C.5 analyzes contamination chronologically to prove robust generalization. However, since UOJ is a fully open-access platform (Section 3.1), isn't it highly probable that pre-training corpora like GitHub already contain these exact UOJ hack submissions and repair patches regardless of the year they were published?

**Limitations:**

1. Economic Bottlenecks: As detailed in the cost analysis (Section 4.3), the massive token consumption of LLM-based judging (estimated at $100,000 annually for UOJ) makes real-world deployment economically unsustainable compared to traditional CPU judging.
2. Restricted Scope of Fixes: Enforcing a maximum 10% modification patch constraint for Task 3 validation (Section 4.2) fundamentally limits the evaluation to localized bug fixes rather than structural algorithmic corrections.
3. Ceiling on Algorithmic Vulnerabilities: The case study (Appendix D.1) demonstrates that current LLMs using brute-force test-time scaling completely fail on structure-dependent algebraic vulnerabilities, indicating a hard capability ceiling for the proposed methods.

**Strengths And Weaknesses:**

Strengths:
1. Novel Task Formulation: The benchmark successfully extends beyond standard generation to evaluate "code hacking" and "repair," providing a much-needed perspective on LLMs as educational debugging assistants (Section 1).
2. High-Quality Real-World Data: By leveraging UOJ's unique community "Hack" mechanism, the authors brilliantly isolate "covert errors" (bugs that pass standard tests but fail edge cases) from overt errors, significantly advancing past prior benchmarks like REFUTE (Sections 3.1 & 3.2).
3. Native Judging Infrastructure: Unlike existing works that exclude non-trivial grading, UOJ-Bench integrates directly with UOJ's internal API, allowing for the rigorous evaluation of custom checkers, Time Limit Exceeded (TLE), and Memory Limit Exceeded (MLE) constraints (Figure 1 & Section 3.1).

Weaknesses:
1. Single-Platform Bias: The dataset relies entirely on a single Chinese online judge (UOJ); Table 8 reveals a minor but consistent performance gap between English and Chinese prompts, suggesting potential language or cultural biases in problem styles (Appendix C.7).
2. Prohibitive Cost of Reproducibility: The authors explicitly note that evaluating test-time scaling costs approximately $1 per problem (Section 4.3), which makes reproducing or extending the Pass@k benchmark results prohibitively expensive for resource-constrained researchers.
3. Heuristic Repair Constraints: The reliance on a strict Levenshtein edit distance ($1 - d_{edit}/L \ge 0.95$) to mine repair pairs might filter out valid, highly educational algorithmic rewrites, artificially narrowing the scope of the repair dataset to only localized typo-level fixes (Section 3.2).

---

> ### Author Rebuttal · Authors · 2026-03-31
>
> ###
>
> We thank the reviewer for the strong score (5) and for highlighting the importance of isolating covert errors, the high-quality real-world data, and our integration with native judging infrastructure.
>
> **W1.1: Single-Platform Bias (UOJ only).**
> While sourced from a single platform, UOJ contains many problems from major international contests (e.g., IOI, APIO) and Chinese national olympiads, which ensures problem diversity. Another reason why we focus on UOJ only is that we can perform native judging through APIs.
>
> **W2 & Lim 1: Prohibitive Cost of Reproducibility.**
> We acknowledge the high cost of brute-force *test-time scaling* ($1/problem) for proprietary models. However, evaluating open-source models (like GPT-OSS-120B) or running the standard Pass@1 benchmark is highly affordable. Given the rapid exponential decrease in API inference costs, we believe this framework establishes a forward-looking capability baseline for when inference costs become feasible.
>
> **W3 & Lim 2: Heuristic repair constraints (Levenshtein distance).**
> To ensure our 0.95 similarity threshold does not artificially narrow the dataset to typo-level fixes, we conducted a robustness check and data distribution analysis. First, our data shows a naturally polarized distribution: among submission pairs with >90% similarity, 85% naturally exceed 95%, and 65% actually exceed 98%. Second, we evaluated the sensitivity of this threshold by increasing it to a strict 0.98. Under this tighter constraint, the repair accuracy of Gemini-3-Pro-Preview was **55.6%**, which is highly consistent with the **53.6%** reported in the paper using the 0.95 threshold. After manual inspection, we found that within this 95% threshold, users successfully implement crucial algorithmic corrections, handle complex edge cases, and fix substantial logic flaws. We set the threshold at 0.95 simply to filter out cases where users gave up and completely rewrote their code from scratch, as complete refactors bypass the targeted "debugging" and "patching" objective of the task.
>
> **Q1: Translation Fidelity.**
> To verify translation quality, we manually inspected a diverse sample of the translated problem statements, including the most complex ones. We found that the LLM translations were excellent: LLMs preserve all critical mathematical constraints, subtle hints, and background stories. The minimal 1.5% performance gap between English and Chinese prompts (Table 8) further validates that critical information was not lost.
>
> **Q2: Zero-Day Verification.**
> Newly discovered bugs are strictly verified by the live UOJ judging system. When a hack is successfully submitted, UOJ automatically triggers a global re-evaluation of all accepted submissions. If the platform's official reference solution breaks (causing almost all accepted solutions to fail), it indicates a flawed problem setup, and UOJ administrators manually intervene. If the hack only breaks a few specific submissions (including our target) while the reference solution stands, it is validated by the system as a true covert bug.
>
> **Q3: Data Contamination Nuance.**
> We must emphasize a critical nuance of the UOJ platform: while problem statements and user submissions are public, the problem data, including **Extra Tests** (the specific input test cases used for hacking) are hidden from the public. Therefore, even if a corpus like GitHub scraped the platform, it would not contain the specific hack inputs. Furthermore, for the Repair dataset, scraping public submissions does not natively pair a buggy submission with its exact fix. Because these precise pairs and hidden test cases are not readily available in unstructured pre-training data, contamination risk is significantly lower (as empirically supported in Appendix C.5).

---

### Official Review · Reviewer_KuCh · 2026-03-16

**Soundness:** 4
**Presentation:** 4
**Significance:** 3
**Originality:** 3
**Overall Recommendation:** 5
**Confidence:** 4

**Summary:**

The authors present a new benchmark which uses problems from a number of different sources that have been aggregated on the UOJ competitive programming platform in China. With integration into the online UOJ grading system, the benchmark promises accurate auto-scoring of generations. The benchmark covers not only generation of complete solutions but also repair of flawed human-written solutions (Repair) and the task of writing test cases to disprove incorrect solutions (Hack).

While exporting past community hacks as the main Hack benchmark, the authors also create the Zero-Day-Hacking-5k corpus of solutions with no known flaws to test whether modern LLMs can discover new, unknown Hacks, and find this is possible on a few of the problems.

Problems span different difficulty levels and present a challenge to modern models, as shown in the authors’ main evaluation suite. In addition to a main model comparison, the authors conduct a number of ablations, demonstrating that:
Test-time scaling of cheaper models is useful and can outperform more expensive models at matched cost budgets
On the repair task, agentic test time scaling is more effective than parallel k-generation, but this is not true for the Hack task
Contamination of Repair + Hack tasks is limited, but contamination of Generation tasks is noticeable.

**Compliance With Llm Reviewing Policy:**

Affirmed.

**Final Justification:**

I was already in favor of acceptance, and my rebuttal concerns were minor. I think the authors have addressed them. I have no open questions and my score is maintained.

**Key Questions For Authors:**

1. In the Key features section, you mention that the “collaboration with UOJ” is what permits access to the “internal API” for testing. I notice that in the downloadable code there is a UOJ API KEY needed for evaluating against the hosted test servers. Will this make it challenging for arbitrary community members to make use of the UOJ tests? Or is the internal API available to anyone for ongoing use of this benchmark?
2. In Appendix C.9, you mention that TLE is harder to repair because of “the intuition that addressing TLE typically requires substantive algorithmic optimization or a rethinking of the overall approach, whereas hacking or repairing RE and WA often involves identifying and exploiting localized issues such as boundary conditions or logic flaws.” Is it possibly the case that this low performance is actually a structural constraint enforced by the <10% edit distance requirement for repair solutions? If relaxed for TLE, does this increase the solve rate? What proportion of solutions are failing this <10% check for TLE?
3. How does the live evaluation of submissions on the UOJ server play with the benchmark? You describe a procedure in which new hacks are applied as test cases to all submissions for the given problem. Does this mean that the results presented in your tables might potentially change as the solutions generated by models are potentially shown to be slightly incorrect by later community-provided hacks?

**Limitations:**

No, the authors do not include a limitations section. I recommend they mention details such as the possible saturation or expiry of the Zero-Day-Hacking-5k challenge as accepted hacks appear on the UOJ site. I also recommend they mention that they include statistics about represented programming languages in solutions–this may be a limitation if the representation of many some languages is small.

**Strengths And Weaknesses:**

Strengths:

1. Very strong and professional presentation
2. Novel benchmark with high utility for increasingly relevant use cases of LLMs (writing unit tests, debugging). Benchmark offers multiple difficulty levels; hardest is not nearly saturated yet.
3. Smart + well-documented design choices for problem curation.
4. Automated evaluation.
5. Useful experiments in main paper + many interesting ablations throughout appendices.
6. Interesting open Zero-Day-Hacking-5k challenge for future LLMs to tackle.

Weaknesses:

1. Little/no discussion of represented programming languages in candidate solutions for hack/repair.
2. Missing a benchmark size comparison against existing work.
3. No refresh/update plan to keep benchmark relevant as current models saturate problems or contamination grows.

Nit: double period at end of abstract. Probably omit the second period after the footnote reference?

---

> ### Author Rebuttal · Authors · 2026-03-31
>
> We sincerely thank the reviewer for the highly positive assessment (Score: 5) and for recognizing the novelty, high utility, and rigorous design choices of UOJ-Bench. We have addressed your constructive feedback below:
>
> **W1 & Limitations: Lack of discussion on programming languages.**
> We acknowledge that our benchmark predominantly features C++. This distribution accurately reflects the reality of competitive programming (CP), where C++ accounts for the vast majority of submissions due to strict execution speed constraints. For comparison, in the existing REFUTE benchmark, 317 out of 324 samples are in C++, with only 7 in Python. We will explicitly state this language distribution in the dataset statistics and note the focus on C++ as a limitation for generalizing to other languages in the final version.
>
> **W2: Missing size comparison against existing work.**
> Thank you for pointing this out. We will add a detailed size comparison in the revision to highlight UOJ-Bench’s scale. For context:
>
> - LiveCodeBench (Original/Pro): 400 / 584 problems
> - CodeELO: 398 problems
> - REFUTE: 324 samples
> - **UOJ-Bench (Ours):** 672 generation problems; 1,525 (479+1046) submissions for Hacking; 716 (500+216) submission pairs for Repair. UOJ-Bench provides a highly competitive volume of strictly curated, human-authored buggy/fixed pairs and adversarial hacks.
>
> **W3: Benchmark refresh/update plan.**
> Because our data curation pipeline (Section 3.2) is highly automated, we plan to release periodic updates for Code Generation and Zero-Day-Hacking (e.g., annually) to UOJ-Bench. By continuously harvesting newly added problems and recent user submissions from the active UOJ platform, we can provide fresh, uncontaminated splits to keep pace with future model capabilities. We will include this maintenance statement in the final version.
>
> **Q1: Will the internal API be available to the community?**
> Yes. We are committed to making the evaluation infrastructure accessible to the research community. Upon publication, we will open-source the full evaluation scripts and set up a process for researchers to request API access. That said, UOJ currently runs on only a couple of CPU servers, so we may need to impose usage limits if demand becomes too high.
>
> **Q2: Is TLE harder to repair because of the <10% edit distance constraint?**
> This is a great question. To investigate, we analyzed the repair failures of Gemini-3-pro-preview. In the Easy track, the ratio of similarity violations for TLE vs. other errors is 10.2% (10/98) vs. 13.2% (53/402). In the Hard track, the ratios are 10.9% (5/46) vs. 4.7% (8/170). These figures show that similarity violations are relatively consistent across error types, suggesting the constraint is not the primary bottleneck for TLE. The true difficulty lies in the nature of the bugs: TLE cases in our dataset arise from fundamentally inefficient algorithms where complexity degrades under specific edge cases. Resolving these requires a holistic rethinking of the algorithm's complexity bottleneck, whereas WA/RE cases often involve localized logic flaws.
>
> **Q3: Live evaluation changing results due to new community hacks?**
> Yes, as the community submits new hacks, the test suite inherently becomes stronger, meaning future scores might slightly decrease as new covert bugs are exposed. However, this dynamic nature is actually a feature. Static test suites (like in many existing benchmarks) are highly susceptible to reward hacking without ever finding the true errors. A dynamic, continuously hardening test suite better reflects real-world software maintenance.
>
> *(Note: The double period and footnote typography errors will be fixed in the revision!)*

---

> > ### Author Rebuttal · Reviewer_KuCh · 2026-04-04
> >
> > Thanks for the rebuttal, I'm satisfied on all points and in favor of acceptance.
> >
> > Re: Q3 (this is not blocking and will not impact my score, but just for the sake of improving the final output), I think what I mean to say is that there should be some kind of ack in the benchmark update plan that whenever scores are computed and exported to a paper or other source, they are only up-to-date as of some moment in time. Perhaps it should be recommended that users of UoJ-Bench report what version or date they used.

---

> > > ### Author Response · Authors · 2026-04-08
> > >
> > > Thank you for your support and for the excellent follow-up suggestion.
> > >
> > > You make a great point about the dynamic nature of the test suite. Since new community hacks are continuously being added, you are exactly right that the evaluation scores represent a snapshot in time. In the final version of the paper and on our repository, we will clearly instruct future users to report the exact date they ran their evaluations. Additionally, we will implement a version number for each individual problem (along with the total number of extra tests), which will be returned via the API call so users can easily track test suite updates.
> > >
> > > Thank you again for helping us improve the reproducibility of the benchmark.

---

### Decision · Program_Chairs · 2026-04-30

**Decision:**

Accept (regular)

**Comment:**

This paper introduces UOJ-Bench, a benchmark designed to evaluate LLMs across three complementary competitive-programming tasks: code generation, code hacking, and code repair. The benchmark is constructed from real-world submissions at UOJ competitive programming platform, and adapts UOJ's internal API for grading. Evaluation of eight frontier models leads to novel insights about the effectiveness/cost of test-time scaling/agentic workflow and the capability of detecting covert errors.

The benchmark is well motivated and built from real-world problems, and with two novel tasks highlighted (hack and repair). Reviewers also found that the automatic evaluation is a preferred property. The distinction between overt and covert errors is interesting. The split by difficulty levels shows that the benchmark is challenging enough for frontier models. On the other hand, reviewers also raised concerns regarding the potential single-platform bias, the inference and judgment cost, and the limitation to the C++ language only. The author's responses clarified most issues raised by the reviewers. All the reviewers support acceptance. Therefore, I recommend it for acceptance.